# Continual Learning by Modeling Intra-Class Variation

**Longhui Yu**                                                                                  *yulonghui@stu.pku.edu.cn*
*School of ECE, Peking University*

**Tianyang Hu, Lanqing Hong***                                                    *{hutianyang1,honglanqing}@huawei.com*
*Huawei Noah's Ark Lab*

**Zhen Liu**                                                                                           *zhen.liu.2@umontreal.ca*
*Mila, Université de Montréal*

**Adrian Weller**                                                                                   *aw665@cam.ac.uk*
*University of Cambridge and The Alan Turing Institute*

**Weiyang Liu***                                                                                   *wl396@cam.ac.uk*
*University of Cambridge and Max Planck Institute for Intelligent Systems - Tübingen*

**Reviewed on OpenReview:** *https://openreview.net/forum?id=iDxfGaMYVr*

## Abstract

It has been observed that neural networks perform poorly when the data or tasks are presented sequentially. Unlike humans, neural networks suffer greatly from catastrophic forgetting, making it impossible to perform life-long learning. To address this issue, memory-based continual learning has been actively studied and stands out as one of the best-performing methods. We examine memory-based continual learning and identify that large variation in the representation space is crucial for avoiding catastrophic forgetting. Motivated by this, we propose to diversify representations by using two types of perturbations: *model-agnostic variation* (*i.e.*, the variation is generated without the knowledge of the learned neural network) and *model-based variation* (*i.e.*, the variation is conditioned on the learned neural network). We demonstrate that enlarging representational variation serves as a general principle to improve continual learning. Finally, we perform empirical studies which demonstrate that our method, as a simple plug-and-play component, can consistently improve a number of memory-based continual learning methods by a large margin.

## 1 Introduction

Recent years have witnessed tremendous success achieved by deep neural networks in a number of applications ranging from object recognition (Krizhevsky et al., 2012) to game playing (Silver et al., 2016). Despite such success, these models stay static once learned and in case of new incoming data, retraining is required, which often suffers from catastrophic forgetting (French, 1999). A naive and highly unscalable solution is to include both old and new data during retraining. Inspired by how humans learn in their lifespan, *continual learning* aims to learn concepts in a sequential and lifelong fashion. In contrast to standard training, continual learning relaxes the *i.i.d.* assumption for training data, which has long been one of the major bottlenecks for existing machine learning models (Hassabis et al., 2017).

The core challenge in continual learning is how to efficiently acquire new knowledge while retaining the old. This problem is closely connected to the *stability-plasticity* dilemma (Ditzler et al., 2015; Grossberg, 1982; Grossberg et al., 2012) in biological systems, where a system should be plastic enough to absorb new knowledge and at the same time stable enough not to catastrophically forget the previous experience. Analogously, the goal of continual learning is to achieve an appropriate trade-off between stability and plasticity in neural network training. Earlier methods to achieve this trade-off can be roughly categorized as regularization-based

---

The code is made publicly available at `https://github.com/yulonghui/MOCA`.

methods (Li & Hoiem, 2017; Rebuffi et al., 2017; Hou et al., 2019; Wu et al., 2019; Yu et al., 2020), dynamic architecture-based methods (Rajasegaran et al., 2019; Hung et al., 2019; Yan et al., 2021) and memory-based methods (Rebuffi et al., 2017; Buzzega et al., 2020; Tiwari et al., 2022; Riemer et al., 2018). Memory-based continual learning preserves learned knowledge by storing a handful of past data points (*i.e.*, prototypes[1]), and is able to achieve competitive performance while only requiring minimal modifications to standard training. Interestingly, memory-based methods also serve as an effective model for approximating the complementary learning systems (McClelland et al., 1995; O'Reilly et al., 2014; O'Reilly & Norman, 2002) where episodic memories are retained by regular experience replay.

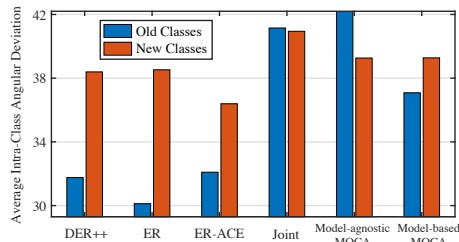

Figure 1: Average intra-class angle deviation (degree) within old classes or new classes. As an example, we use Gaussian perturbation for model-agnostic MOCA and adversarial representational perturbation for modal-based MOCA. All MOCA variants are applied to ER (Riemer et al., 2018). This is computed by final models.

However, approximating the original training data distribution with a few prototypes inevitably results in a lack of intra-class diversity. Figure 1 compares the angle variation of intra-class features (*i.e.*, the average angle between intra-class features and their corresponding class mean) between some popular continual learning method, joint training and our methods, validating that old classes are substantially less diverse in continual learning. Due to the lack of intra-class diversity, we identify two phenomena in memory-based continual learning: *representation collapse* and *gradient collapse*, which lead to poor generalization on past data. Representation collapse corresponds to the phenomenon where the representations from old classes shrink to a straight line during training (*i.e.*, it can be viewed as a point projected on a hypersphere). It happens due to severe overfitting to a few past training data. This is also theoretically grounded in the concept of neural collapse (Papyan et al., 2020). Figure 2 gives a 2D feature

demonstration for representation collapse. Gradient collapse, the phenomenon that the gradients of data collapse in a few directions, is a direct consequence of representation collapse since the direction of the representative determines that of its gradient. We empirically verify the problem of degenerated gradients in Figure 3 by showing that gradients *w.r.t.* representations from old classes are intrinsically low-dimensional and contain limited information. This observation motivates us to increase the intra-class diversity during training to alleviate these two detrimental phenomena and induce regularization that prevents overfitting to past data. To this end, we propose a simple yet effective framework, called MOCA, which models the intra-class variation in the representation space with only a few prototypes. We emphasize that MOCA is quite different from standard data augmentation, since we perform perturbation in the representation space rather than in the input data space. Performing perturbation in the representation space shares a similar spirit to Manifold mixup (Verma et al., 2019).

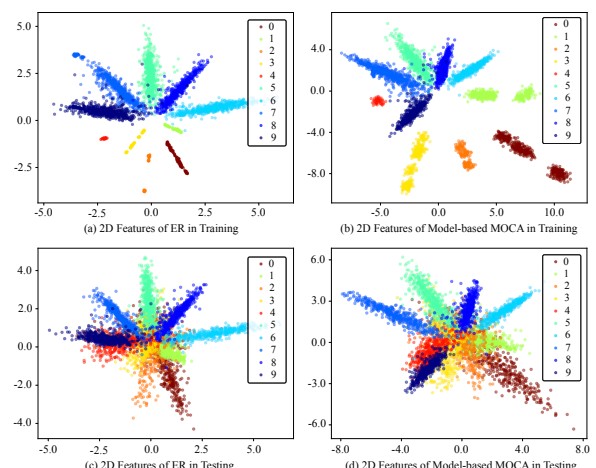

Figure 2: 2D feature visualization of ER (Riemer et al., 2018) (a,c) and ER with model-based MOCA (WAP) (b,d). We construct a simple continual learning task on MNIST, where the first 5 classes are old classes, and the other 5 classes are new classes. We can observe MOCA indeed enlarges the training feature variation, and the testing features generated by MOCA are more discriminative and well separated.

Another motivation behind MOCA comes from our intuition that better mimicking of the gradients in standard *i.i.d.* training leads to better generalization in continual learning. By looking into how the original *i.i.d.* gradients are computed, we find that feature dynamics (*i.e.*, representation at different iterations) and labels can largely determine the back-propagation gradients. If we can approximate or even recover the feature dynamics of *i.i.d.* joint training in the continual learning setting, catastrophic forgetting can be greatly alleviated. However, this is an intrinsically difficult task, since the feature dynamics of standard training are high-dimensional, model-dependent and non-stationary.

---

[1]We use prototypes to represent the raw data points in the paper, which differs from few-shot learning (Snell et al., 2017).

MOCA takes one step closer to this goal by explicitly modeling and effectively enlarging intra-class representation variation. We design two types of intra-class modeling methods: *model-agnostic* MOCA and *model-based* MOCA. Both MOCA variants aim to diversify the intra-class representation variation. Specifically, model-agnostic MOCA diversifies the representation by modeling the intra-class variation with a generic parametric distribution (*e.g.*, Gaussian distribution), and model-based MOCA takes the previously learned model (*i.e.*, a neural network that is trained with the old data) into consideration when diversifying the intra-class representation. For model-agnostic MOCA, we consider Gaussian distribution and von Mises–Fisher (vMF) distribution

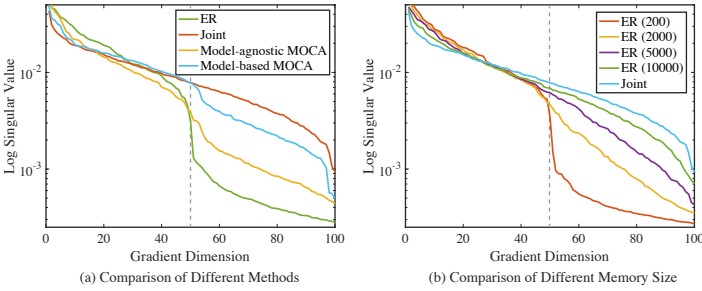

Figure 3: Singular values of (a) the training gradients for different methods (MOCA is trained on top of ER), and (b) the training gradients of ER (Riemer et al., 2018) under different memory size (200, 2000, 5000, 10000). We construct a simple continual learning task for CIFAR-100, where 50 classes are old classes, and the other 50 classes are new classes. We can observe in (a) that both model-agnostic MOCA (Gaussian) and model-based MOCA (WAP) produce gradients with richer directions, and in (b) that larger memory size leads to more informative and diverse gradients, approaching to joint training. Full results are in Appendix D.

for modeling the intra-class variation; for model-based MOCA, we model the intra-class variation by a perturbation dependent on the trained network parameters. To generate such a perturbation, we propose three different methods: Dropout-based augmentation (DOA), weight-adversarial perturbation (WAP) and variation transfer (VT). DOA, inspired by Dropout (Srivastava et al., 2014), augments the data by randomly masking the neurons in a neural net and then viewing the resulting representation as the intra-class perturbation. While DOA perturbs neurons via random masking, WAP perturbs the neurons in an adversarial fashion. Different from DOA and WAP, VT assumes that intra-class variations are similar across different classes and models the variation in old classes with the variations in new classes. Our contributions are listed below:

- Driven by the goal of approximating the gradients of *i.i.d.* joint training, we propose MOCA, a simple yet effective framework to model intra-class variation for continual learning, where both model-agnostic and model-based variations are extensively studied.

- For model-agnostic MOCA, we propose two variants: Gaussian modeling and vMF modeling. For model-based MOCA, we propose three variants: Dropout-based augmentation, variation transfer and weight-adversarial perturbation. Each variant has different modeling assumption and flexibility.

- MOCA serves as a plug-and-play method for memory-based continual learning, which effortlessly improves the empirical performance under offline, online and proxy-based settings. We empirically investigate the performance of MOCA and provide guidance on which variant is likely to work well in different settings.

## 2 Related Work

**Regularization-based approaches** aims to acquire new knowledge while penalizing one of the following three entities: the previously learned parameters, gradient directions and activation outputs. The basic idea of parameter-based regularization method is to resist the change of the important parameters of the learned model. Hence, some recent work (Kirkpatrick et al., 2017; Aljundi et al., 2018; Zenke et al., 2017; Benzing, 2022) explores different measurements of the parameter importance. Similar to our work, gradient-based regularization methods (Lopez-Paz & Ranzato, 2017; Chaudhry et al., 2018; Saha et al., 2021) try to restrict the update directions in a common space where gradients from old classes and new classes have the largest inner product. Some of these methods also aim to separate the network parameters for old tasks and new tasks by adjusting training gradients. For instance, Farajtabar et al. (2020) project the new training gradients to the orthogonal space of the old training gradients to minimize the change in neural activations for old tasks. Similarly, Adam-NSCL (Wang et al., 2021) first finds the null space for old tasks by analyzing the covariance matrix of all the input features for each layer and then projects the gradients into the null space to prevent forgetting. The last type of method, activation-based regularization (Li & Hoiem, 2017; Rebuffi et al., 2017; Hou et al., 2019; Wu et al., 2019; Yu et al., 2020), leverages the information from the activations obtained from past tasks using strategies similar to knowledge distillation (Hinton et al., 2015). A notable

example is iCaRL (Rebuffi et al., 2017), which uses knowledge distillation to transfer the old knowledge in the memory buffer by herding (Welling, 2009) from the previous model to the learned model. A number of methods (Wu et al., 2019; Hou et al., 2019; Liu et al., 2020; Douillard et al., 2020) take a path similar to iCaRL and utilize an extra model with a memory buffer to prevent forgetting. DER (Buzzega et al., 2020) improves iCaRL by preserving the old logit activations rather than ground truth labels with a distilled memory buffer. Additionally, Mirzadeh et al. (2020) studies how different training regimes (*e.g.*, learning rate, batch size) can affect the continual learning performance. Mirzadeh et al. (2022) discusses the impact of architectures on continual learning. Different from regularization-based methods, we focus on the problem of representation and gradient collapse and address it by modeling intra-class variation of old classes.

**Dynamic-architecture-based approaches.** Similar to regularization-based approaches that aim to separate network parameters, gradients and activations for both old and new classes, Dynamic-architecture-based approaches aim to directly separate the network parameters into subsets of task-specific ones. For instance, PNN (Rusu et al., 2016) freezes the parameters trained on old tasks but introduces new trainable sub-networks to the existing trained network to adapt to new tasks. In addition to the simple network expansion, Rajasegaran et al. (2019); Hung et al. (2019); Cao et al. (2022) perform additional steps to freeze or prune the old network parameters to prevent forgetting, while HAT (Serra et al., 2018) utilizes a hard attention mask to constrain the amount of updates of important neurons. DynamicalER (Yan et al., 2021) takes a slightly different approach by sequentially training independent networks for each task. Then the learned representations are also used to encourage the difference between the representations of old and new tasks. von Oswald et al. (2020) uses task-conditioned hypernetworks to generate network weights. Different from the dynamic-architecture-based approaches that introduce new network structures for new tasks, our paper aims to prevent catastrophic forgetting in a parameter-efficient and scalable way without any significant modification to the existing neural architecture.

**Memory-based approaches** maintain a buffer storing a small subset of past data to improve continual learning and achieve appealing performance in both offline and online continual learning. During training, both the buffer data and the incoming new data are included in the mini-batches. Due to the memory constraint of the replay buffer, it is important to design a good sample selection strategy so that the buffer stores the most representative data to prevent forgetting (Wang et al., 2022). Experience Replay (ER) (Riemer et al., 2018) establishes a simple baseline by storing randomly selected subsets of data into the memory buffer. MIR (Aljundi et al., 2019a) selects the data whose losses are most sensitive to the data of the next task. GSS (Aljundi et al., 2019b) proposes to build memory buffers with the largest sample diversity and gradient variance. ASER (Shim et al., 2021) leverages Shapley value (Roth, 1988) to select buffer data. GCR (Tiwari et al., 2022) proposes a gradient-based selection strategy by approximating the gradients of all the data seen so far with respect to current model parameters. With memory buffers, regularization-based methods can better utilize the old activation knowledge (Rebuffi et al., 2017; Buzzega et al., 2020) and old gradient information (Lopez-Paz & Ranzato, 2017; Wang et al., 2021; Farajtabar et al., 2020). Instead of storing old samples, DGR (Shin et al., 2017) trains a generative model (Goodfellow et al., 2014a) to synthesize old data.

Our work is mostly related to the memory-based approaches. Existing memory-based approaches focus on either regularizing the old knowledge (Buzzega et al., 2020; Rebuffi et al., 2017; Caccia et al., 2021) or sampling the most representative data points (Riemer et al., 2018; Aljundi et al., 2018; 2019b). Taking a different perspective, we identify a general principle to improve continual learning – bridging the large variation gap between the representations of the buffer data (*i.e.*, old data) and new data. We discover that such a gap results in representation and gradient collapse which is harmful to generalization in continual learning. Motivated by this observation, MOCA models the intra-class representation variation such that the representation and gradient variation of the data in the buffer can match the *i.i.d.* joint training scenario.

## 3   Why Can Modeling Intra-class Variation Help Continual Learning?

**Preliminaries**. In this section, we briefly present the background knowledge for memory-based offline continual learning, proxy-based continual learning and memory-based online continual learning. Let $\mathcal{T}_1$, $\mathcal{T}_2, \ldots, \mathcal{T}_t$ represent the sequence of continual learning tasks. Each task has *i.i.d.* samples from the task data distribution $\mathcal{D}^t$, and the composed training dataset is denoted by $(\boldsymbol{x}^t, y^t) \sim \mathcal{D}^t$, where $\boldsymbol{x}^t$ is the input data and $y^t$ is the label. The new classes in the $t$-th task $\mathcal{T}_t$ is denoted by $\mathcal{C}_t = \{c_{k_{t-1}+1}, c_{k_{t-1}+2}, \ldots, c_{k_t}\}$, a set of

($k_t$ - $k_{t-1}$) classes. We aim to find a model, comprised of a feature extractor and a top classifier, that can perform well among all the learned tasks. More specifically, we consider a neural network $h_{\theta}$ parameterized by $\theta$, the output of which $h_{\theta}(x)$ is the feature representation of $x$ to be fed to the classifier layer $g_{\phi}(\cdot)$ (parameterized by $\phi$). $g_{\phi}(h_{\theta}(x))$ is a vector representing the predicted confidence of $x$ for the total $k$ classes. $k$ is the total number of classes till the task $\mathcal{T}_t$. The general objective of continual learning is:

$$\min_{\theta,\phi} \sum_{t=1}^{\mathcal{T}_t} \mathbb{E}_{(x^t,y^t)\sim\mathcal{D}^t}[\mathcal{L}(g_{\phi}(h_{\theta}(x^t)), y^t)], \qquad (1)$$

where $\mathcal{L}$ is some loss function for classification. Usually, $\mathcal{L}$ is chosen to be the cross-entropy between $g_{\phi}(h_{\theta}(x))$ and $e_y$, formally written as $\mathcal{L}_{ce}(g_{\phi}(h_{\theta}(x)), e_y) = -\log \frac{\exp(g_{\phi}(h_{\theta}(x))_y)}{\sum_i \exp(g_{\phi}(h_{\theta}(x))_i)}$.

Under the continual learning setting, the old task data $\{(x^t, y^t) \sim \mathcal{D}^t : t = 1, \ldots, t-1\}$ are mostly unavailable while learning the current task $\mathcal{T}_t$. The lack of the old class data causes overfitting to task $\mathcal{T}_t$ and catastrophic forgetting of previous knowledge. To prevent catastrophic forgetting, memory-based offline continual learning (*e.g.*, (Buzzega et al., 2020)) usually preserves a replay memory buffer $(x^{old}, y^{old}) \in \mathcal{M}$ of limited size and optimizes the following general training objective:

$$\min_{\theta,\phi} \mathbb{E}_{(x^t,y^t)\sim\mathcal{D}^t}[\mathcal{L}_{ce}(g_{\phi}(h_{\theta}(x^t)), y^t)] + \mathbb{E}_{(x^{old},y^{old})\in\mathcal{M}}[\mathcal{L}_{ce}(g_{\phi}(h_{\theta}(x^{old})), y^{old})]. \qquad (2)$$

In proxy-based continual learning (*e.g.*, (Zhu et al., 2021)), we consider the memory-free scenario where old task data $x^{old}$ can not be stored. Instead of the old task data $x^{old}$, the mean representation $\bar{f}_i$ for each class $i \in \{c_1, c_2, \ldots, c_{k_{t-1}}\}$ is stored and reusable. In this case, the general training objective is:

$$\min_{\theta,\phi} \mathbb{E}_{(x^t,y^t)\sim\mathcal{D}^t}[\mathcal{L}_{ce}(g_{\phi}(h_{\theta}(x^t)), y^t)] + \sum_{i=1}^{k_{t-1}} \mathbb{E}[\mathcal{L}_{ce}(g_{\phi}(\bar{f}_i), y^i)]. \qquad (3)$$

Memory-based online continual learning (*e.g.*, (Caccia et al., 2021)) is a more memory-friendly setting, which has the same training objective as Equation 2 but all the training data can only be sampled and used once.

### 3.1  On Approximating Joint Training Gradients with Intra-class Representation Modeling

We start by analyzing the back-propagated gradients *w.r.t.* the representation $h_{\theta}(x)$ and obtain that

$$\frac{\partial \mathcal{L}_{ce}}{\partial h_{\theta}(x)} = \mathbb{E}_{(x,y)\sim\mathcal{D}}\left(\text{Softmax}(g_{\phi}(h_{\theta}(x))) - e_y\right) \cdot \frac{\partial g_{\phi}(h_{\theta}(x)))}{\partial h_{\theta}(x)},$$

$$= \mathbb{E}_{(f,y)\sim\tilde{\mathcal{D}}}\left(\text{Softmax}(g_{\phi}(f)) - e_y\right) \cdot \frac{\partial g_{\phi}(f)}{\partial f}, \qquad (4)$$

(a) Original  (b) Model-agnostic MOCA  (b) Model-based MOCA

Figure 4: Graphical models of intra-class representation modeling for (a) original memory-based continual learning and (b,c) the MOCA framework.

where $\text{Softmax}(v) = \left\{\frac{\exp(v_1)}{\sum_{i=1}^d \exp(v_i)}, \cdots, \frac{\exp(v_d)}{\sum_{i=1}^d \exp(v_i)}\right\} \in \mathbb{R}^{1\times d}$ ($v \in \mathbb{R}^{1\times d}$ is a $d$-dimensional vector) and $f$ denotes the feature representation of $x$ (*i.e.*, $f = h_{\theta}(x)$). From Equation 4, we can observe that the back-propagated gradients for updating the neural network $h_{\theta}$ is uniquely determined by $f$, $y$ and $\phi$. $\phi$ is typically parameterized as a linear classifier which is easy to compute given $f$ or can be well approximated with moving-averaged class centroids (Wen et al., 2019). Therefore the problem of approximating the original gradients, to large extent, reduces to modeling the intra-class representation, *i.e.*, finding an approximate $f^y \sim \tilde{\mathcal{D}}(y, \theta)$ where $\tilde{\mathcal{D}}(y, \theta)$ denotes the feature distribution of the $y$-th class for the model $\theta$. One may notice that to approximate the gradients, we could either approximate the distribution of $f$ or the distribution of $x$. We seek to approximate the distribution of $f$ rather than the distribution of $x$ because the latent space produced by neural networks is more regularized and feature distributions for different classes also tend to be similar (see Figure 2 as an example). In contrast, modeling the distribution of $x$ is essentially to build a generative model for raw images, which itself is a highly challenging task especially with only a limited amount of images available. The construction of the memory buffer (Rebuffi et al., 2017) is essentially to approximate the distribution of

$\boldsymbol{x}$ with a few representative examples. Taking advantage of the memory buffer, we instead propose to model the intra-class representation with $\boldsymbol{f}^y = \boldsymbol{f}_M^y + \Delta\boldsymbol{f}^y$ where $\boldsymbol{f}_M^y$ denotes the prototypes from the $y$-th class in the memory buffer (*i.e.*, $\boldsymbol{f}_M^y \in \{h_{\boldsymbol{\theta}}(\boldsymbol{x}_1^y), \cdots, h_{\boldsymbol{\theta}}(\boldsymbol{x}_m^y)\}$ where $\boldsymbol{x}_i^y$ is the $i$-th prototype of the $y$-th class) and $\Delta\boldsymbol{f}^y$ is the deviation between the actual representation and the prototype for the $y$-th class. In other words, original memory-based continual learning approximates the distribution of $\boldsymbol{f}$ only with prototypes $\boldsymbol{f}_M$ from the memory buffer, while MOCA additionally approximates the distribution of $\Delta\boldsymbol{f}$ with either a generic distribution or a model-based variation, as shown in Figure 4.

## 3.2 Why Is Modeling Representation Better Than Modeling Raw Input Images?

To simulate intra-class variation of features in *i.i.d.* training, Equation 4 suggests that we can model either the distribution of the raw input images $\boldsymbol{x}$ or the distribution of the representation $\boldsymbol{f}$. We argue that modeling the intra-class representation variation is much easier than modeling raw input images based on the following reasons. First, the dimensionality of the representation space is usually much lower than that of the raw input images, making it easier to model the intra-class variation. Second, the representation space is more regularized, since it converges to the simplex equiangular tight frame (Papyan et al., 2020) which is also equivalent to a hyperspherically uniform space (Liu et al., 2018a; Lin et al., 2020; Liu et al., 2021c;b). Moreover, we empirically observe from Liu et al. (2016; 2018b); Wen et al. (2019) that the intra-class representations, when projected onto a unit hypersphere, are centered around the class mean and distributed like a vMF distribution. This observation directly motivates us to model the intra-class variation with a parametric distribution in the representation space, leading to model-agnostic MOCA. Last, the features from different classes share similar hyperspherical variation in the representation space (empirically validated by Liu et al. (2016)), while variation in the raw image space is completely different for different classes.

## 3.3 Modeling Intra-Class Representation Variation as Implicit Data Augmentation

Modeling intra-class representation variation can implicitly serve as a form of data augmentation. Since we aim to model the distribution of $\Delta\boldsymbol{f}^y$ in MOCA, the resulting back-propagated gradient is computed by the feature $\boldsymbol{f}_M^y + \Delta\boldsymbol{f}^y$. This new gradient can be viewed as being generated by a augmented input data $\tilde{\boldsymbol{x}}^*$:

$$\tilde{\boldsymbol{x}}^* := \arg\min_{\tilde{\boldsymbol{x}}} \|h_{\boldsymbol{\theta}}(\tilde{\boldsymbol{x}}) - \boldsymbol{f}_M^y - \Delta\boldsymbol{f}^y\|_F^2 = \arg\min_{\tilde{\boldsymbol{x}}} \|h_{\boldsymbol{\theta}}(\tilde{\boldsymbol{x}}) - h_{\boldsymbol{\theta}}(\boldsymbol{x}) - \Delta\boldsymbol{f}^y\|_F^2 , \tag{5}$$

where $\tilde{\boldsymbol{x}}^*$ can generate the same gradient as the perturbed representation $h_{\boldsymbol{\theta}}(\boldsymbol{x}) + \Delta\boldsymbol{f}^y$ once the minimization can attain zero ($\boldsymbol{x}$ denotes a prototype from the $y$-th class). There are multiple solutions for the augmented data $\tilde{\boldsymbol{x}}$ given different $\boldsymbol{x}$ and $\boldsymbol{\theta}$. Even for the same set of $\boldsymbol{x}$ and $\boldsymbol{\theta}$, $\tilde{\boldsymbol{x}}$ will also have different solutions due to the highly non-convex nature of neural networks, but all these solutions lead to the gradient as induced by the same $\Delta\boldsymbol{f}$. Therefore, MOCA can be viewed as generating numerous equivalent augmented data at the same time by perturbing the representation space, which is quiet different from explicit data augmentation. This many-to-

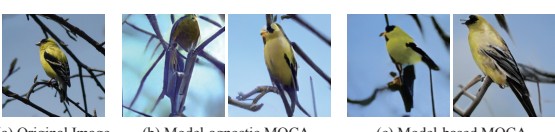

(a) Original Image  (b) Model-agnostic MOCA  (c) Model-based MOCA

Figure 5: Some implicit augmented images for the old class "bird". For model-agnostic MOCA, we use normalized Gaussian distribution. For model-based MOCA, we use WAP. The augmented examples are generated by a pretrained generative model. We ensure that these augmented images produce the same feature representation as MOCA, so they share the same back-propagated gradient. Detailed visualization procedure is given in Appendix B.

one mapping property of neural networks is one of the reasons that modeling intra-class variation in the representation space is easier than modeling in the raw data space. The same intuition has also been adopted in natural language processing when the raw data (*e.g.*, sentence) is hard to augment (Gao et al., 2021). With the implicit data augmentation induced by MOCA, representation collapse can be greatly alleviated.

# 4 ☕ MOCA: A Framework for Modeling Intra-Class Variation

## 4.1 Framework Overview

Aiming to model intra-class variation in the representation space, MOCA produces augmented representations with $\boldsymbol{f} = h_{\boldsymbol{\theta}}(\boldsymbol{x}) + \Delta\boldsymbol{f}$ where $\boldsymbol{x}$ denotes a prototype from the $y$-th class and $\Delta\boldsymbol{f}$ is the deviation added by MOCA. For better modeling expressiveness, there could be multiple prototypes per old class. Model-agnostic

MOCA generates $\Delta \boldsymbol{f}$ using a parametric distribution without the knowledge of the model $\boldsymbol{\theta}$, and model-based MOCA generates $\Delta \boldsymbol{f}$ by taking the model $\boldsymbol{\theta}$ into consideration.

**Modeling hyperspherical variation.** To enable an effective modeling of $\Delta \boldsymbol{f}$, we draw inspirations from the observations in (Liu et al., 2017b; 2018b;a; Chen et al., 2020) that geodesic distance on hypersphere is well aligned with perceptual difference, and propose to model the distribution $\Delta \boldsymbol{f}$ on the hypersphere in MOCA. We argue that projection on hypersphere can limit the space of $\Delta \boldsymbol{f}$ to a semantically meaningful one. To this end, we additionally perform a projection step on $\boldsymbol{f}$ to ensure that its magnitude is the same as $h_{\boldsymbol{\theta}}(\boldsymbol{x})$, yielding the final augmented

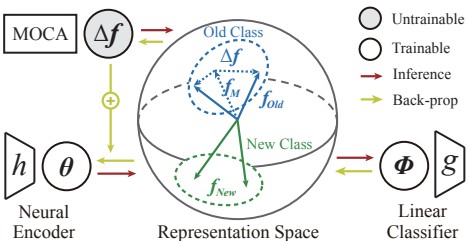

Figure 6: Inference and back-prop in MOCA.

feature as $\boldsymbol{f} = \frac{\|h_{\boldsymbol{\theta}}(\boldsymbol{x})\|}{\|h_{\boldsymbol{\theta}}(\boldsymbol{x})+\tilde{\Delta}\boldsymbol{f}\|}(h_{\boldsymbol{\theta}}(\boldsymbol{x}) + \tilde{\Delta}\boldsymbol{f})$ where $\tilde{\Delta}\boldsymbol{f}$ is a unconstrained perturbation. We then end up with the following formulation for the augmented feature on the prototype feature hypersphere:

$$\underbrace{\boldsymbol{f}}_{\text{Augmented Feature}} = \underbrace{h_{\boldsymbol{\theta}}(\boldsymbol{x})}_{\text{Prototype Feature}} + \underbrace{\left( \big( \|h_{\boldsymbol{\theta}}(\boldsymbol{x})\| - \|h_{\boldsymbol{\theta}}(\boldsymbol{x}) + \tilde{\Delta}\boldsymbol{f}\| \big) h_{\boldsymbol{\theta}}(\boldsymbol{x}) + \|h_{\boldsymbol{\theta}}(\boldsymbol{x})\| \tilde{\Delta}\boldsymbol{f} \right) \big\|h_{\boldsymbol{\theta}}(\boldsymbol{x}) + \tilde{\Delta}\boldsymbol{f}\big\|^{-1}}_{\text{Hyperspherical Augmentation } \Delta \boldsymbol{f}},$$

where $\Delta \boldsymbol{f}$ denotes the perturbation on the hypersphere (*i.e.*, angular perturbation) and it does not change the magnitude of the original prototype feature $h_{\boldsymbol{\theta}}(\boldsymbol{x})$ (*i.e.*, $\|\boldsymbol{f}\| = \|h_{\boldsymbol{\theta}}(\boldsymbol{x})\|$). With the hypersphere constraint, MOCA reduces the difficulty of finding a good $\Delta \boldsymbol{f}$. Moreover, such a design implicitly constrains the intra-class representation modeling to be semantic (Liu et al., 2018b; Chen et al., 2020).

**Back-propagated gradient in MOCA.** Taking advantage of the augmented feature, we design the back-propagated gradient *w.r.t.* the representation to be $\frac{\partial \mathcal{L}_{ce}}{\partial \boldsymbol{f}} = \frac{\partial \mathcal{L}_{ce}}{\partial h_{\boldsymbol{\theta}}(\boldsymbol{x})} + \frac{\partial \mathcal{L}_{ce}}{\partial \Delta \boldsymbol{f}}$ instead of the original $\frac{\partial \mathcal{L}_{ce}}{\partial h_{\boldsymbol{\theta}}(\boldsymbol{x})}$, as illustrated in Figure 6. This gradient is used to update the parameters $\boldsymbol{\theta}$ of the encoder $h(\cdot)$. From the back-propagation perspective, MOCA can also be viewed as a gradient augmentation method. We show that the augmented gradient can well prevent both representation collapse and gradient collapse, leading to a discriminative feature representation in continual learning.

**Usefulness of the model for intra-class variation.** To model the unconstrained perturbation $\tilde{\Delta}\boldsymbol{f}$, we can use a parametric distribution such as Gaussian distribution and vMF distribution. This leads to a simple variant: model-agnostic MOCA where $\tilde{\Delta}\boldsymbol{f}$ does not depend on the current model $\boldsymbol{\theta}$. However, an accurate modeling of $\tilde{\Delta}\boldsymbol{f}$ should take the model $\boldsymbol{\theta}$ into consideration, because the representation $\boldsymbol{f} = h_{\boldsymbol{\theta}}(\boldsymbol{x})$ is always conditioned on $\boldsymbol{\theta}$. To leverage the current model parameters $\boldsymbol{\theta}$, we come up with an advanced variant: model-based MOCA where $\tilde{\Delta}\boldsymbol{f}$ is generated based on $\boldsymbol{\theta}$. Our experiments comprehensively validate the effectiveness of both model-agnostic MOCA and model-based MOCA.

## 4.2 Model-agnostic MOCA

For model-agnostic MOCA, we propose to use two simple distributions: Gaussian distribution and vMF distribution to model the intra-class variation in the representation space.

**Isotropic Gaussian distribution.** To enlarge the variation of old class feature space, we propose to model $\tilde{\Delta}\boldsymbol{f}$ with an isotropic Gaussian distribution. We denote the prototype feature (from old classes) produced by the neural network as $h_{\boldsymbol{\theta}}(\boldsymbol{x}^{\text{old}})$ where $\boldsymbol{x}^{\text{old}}$ is the stored prototype for old classes. This model-agnostic MOCA variant is named as Gaussian for simplicity and the final perturbed feature $\boldsymbol{f}$ can be written as

$$\boldsymbol{f} = \big\|h_{\boldsymbol{\theta}}(\boldsymbol{x}^{\text{old}})\big\| \cdot \mathcal{P}_{\mathbb{S}}\big(\mathcal{P}_{\mathbb{S}}(h_{\boldsymbol{\theta}}(\boldsymbol{x}^{\text{old}})) + \lambda \cdot \boldsymbol{\epsilon}\big), \tag{6}$$

where $\boldsymbol{\epsilon} \sim \mathcal{N}(0, \boldsymbol{I})$ is the isotropic Gaussian noise with the same dimension as the original old-class feature, $\mathcal{P}_{\mathbb{S}}(\boldsymbol{v}_0)$ denotes the projection operator onto the unit hypersphere that outputs $\arg\min_{\boldsymbol{v}\in\mathbb{S}^d} \|\boldsymbol{v} - \boldsymbol{v}_0\|_2^2$ (usually we have that $\mathcal{P}_{\mathbb{S}}(\boldsymbol{v}_0) = \frac{\boldsymbol{v}_0}{\|\boldsymbol{v}_0\|}$), and $\lambda$ denotes the variance and also controls the perturbation magnitude. The inner projection that applies to $h_{\boldsymbol{\theta}}(\boldsymbol{x}^{\text{old}})$ ensures the consistency of $\lambda$ for different prototypes.

**von Mises–Fisher distribution.** Since we are modeling intra-class variation on a hypersphere, the von Mises–Fisher (vMF) distribution appears to be a valid choice. The vMF distribution is parameterized by a

mean direction $\boldsymbol{\mu}$ and a concentration parameter $\kappa$. We name this model-agnostic MOCA variant as vMF and we can produce the perturbation with arbitrary angle between the original prototype feature by adjusting the concentration parameter $\kappa$. The augmented feature is written as $\boldsymbol{f} = \|h_{\boldsymbol{\theta}}(\boldsymbol{x}^{\text{old}})\| \cdot \mathcal{P}_{\mathbb{S}}(\mathcal{P}_{\mathbb{S}}(h_{\boldsymbol{\theta}}(\boldsymbol{x}^{\text{old}})) + \lambda \cdot \boldsymbol{\epsilon})$, where the random variable $\boldsymbol{\epsilon}$ follows the probability density function shown below:

$$p(\boldsymbol{\epsilon}|\boldsymbol{\mu}, \kappa) = \frac{\kappa^{d/2-1}}{(2\pi)^{d/2}I_{d/2-1}(\kappa)} \exp(\kappa\boldsymbol{\mu}^{\top}\boldsymbol{\epsilon}), \quad \boldsymbol{\mu} = \mathcal{P}_{\mathbb{S}}\big(h_{\boldsymbol{\theta}}(\boldsymbol{x}^{\text{old}})\big), \tag{7}$$

where $I_v$ denotes the modified Bessel function of the first kind at order $v$ and $d$ is the dimension of the feature space. The vMF distribution becomes more concentrated with larger $\kappa$. When $\kappa = 0$, the vMF distribution reduces to uniform distribution on the hypersphere. The sampling of the vMF distribution follows the procedure of Ulrich (1984); Davidson et al. (2018) and the detailed algorithm is given in Appendix A.

## 4.3 Model-based MOCA

Model-agnostic MOCA can diversify the intra-class distribution of old classes using a generic parametric distribution without considering the specific model parameters. Therefore, model-agnostic MOCA has to treat all the feature dimension equally, which might not match the intrinsic feature distribution. To fill up this gap, we further consider model-based MOCA where the feature augmenta-

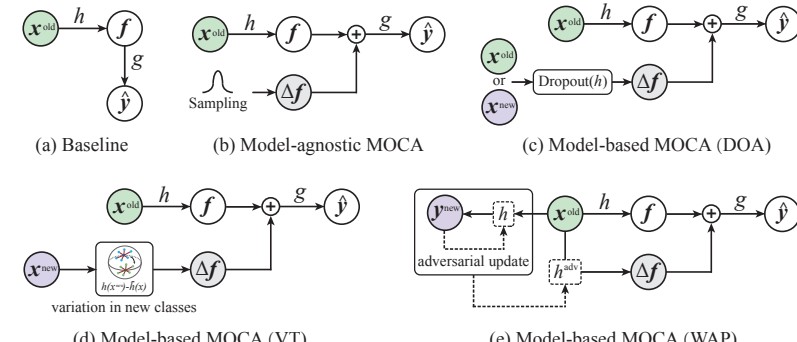

Figure 7: Illustration of different MOCA variants.

tion is generated based on the current model parameters. By considering the model's knowledge, model-based MOCA can augment the feature with informative directions (unlike the isotropic distribution used in model-agnostic MOCA). To this end, the basic idea behind model-based MOCA is to generate the augmentation $\tilde{\Delta}\boldsymbol{f}$ by perturbing the encoder $h_{\boldsymbol{\theta}}(\boldsymbol{x})$. There are generally two ways to perturb $h_{\boldsymbol{\theta}}(\boldsymbol{x})$ – perturbing either the model parameters $\boldsymbol{\theta}$ or the input $\boldsymbol{x}$. Conceptually, we have the following general feature perturbation models:

$$\text{Perturbation I: } \tilde{\Delta}\boldsymbol{f} = \lambda_1 h_{\boldsymbol{\theta}}(\boldsymbol{x}) - \lambda_2 h_{\boldsymbol{\theta}+\Delta\boldsymbol{\theta}}(\boldsymbol{x}), \quad \text{Perturbation II: } \tilde{\Delta}\boldsymbol{f} = \lambda_1 h_{\boldsymbol{\theta}}(\boldsymbol{x}) - \lambda_2 h_{\boldsymbol{\theta}}(\boldsymbol{x} + \Delta\boldsymbol{x}), \tag{8}$$

where $\boldsymbol{x}$ could be samples from old classes or new classes (depending on the specific continual learning setting). We can observe that both perturbation models make the feature augmentation $\tilde{\Delta}\boldsymbol{f}$ dependent on the model parameters $\boldsymbol{\theta}$. For the first perturbation model, we derive two instances: Dropout-based augmentation and weight-adversarial perturbation. Inspired by recent progresses in contrastive learning of natural language (Gao et al., 2021; Liu et al., 2021a), DOA uses Dropout (Srivastava et al., 2014) to generate $\Delta\boldsymbol{\theta}$, which is essentially to randomly mask out neurons. Different from DOA, WAP uses adversarial training (Szegedy et al., 2013) to generate $\Delta\boldsymbol{\theta}$ such that the resulting feature becomes closer to the decision boundary. For the second perturbation model, we make use of the accessible variation in the new class and propose to directly transfer the feature variation to the old classes, leading to a variant called variation transfer.

**Dropout-based augmentation.** Dropout (Srivastava et al., 2014) was initially used for regularizing neural networks to avoid overfitting. Recently, it has been discovered that Dropout can serve as a form of data augmentation (Gao et al., 2021). Inspired by this, we utilize Dropout to diversify the representation distributions for old classes. Specifically, we have that $\tilde{\Delta}\boldsymbol{f} = \lambda_1 h_{\boldsymbol{\theta}}(\boldsymbol{x}) - \lambda_2 h_{\text{Dropout}(\boldsymbol{\theta})}(\boldsymbol{x})$ where $\text{Dropout}(\boldsymbol{\theta})$ denotes the Dropout operator applied on the network $\boldsymbol{\theta}$ and $\lambda_1, \lambda_2$ are hyperparameters (In this paper, $\lambda_1 = \lambda_2 = 1$). After simplifying hyperparameters and hyperspherical projection, we can write DOA as

$$\boldsymbol{f} = \|h_{\boldsymbol{\theta}}(\boldsymbol{x}^{\text{old}})\| \cdot \mathcal{P}_{\mathbb{S}}\left(\mathcal{P}_{\mathbb{S}}\left(h_{\boldsymbol{\theta}}\left(\boldsymbol{x}^{\text{old}}\right)\right) + \lambda \cdot \mathcal{P}_{\mathbb{S}}\left(h_{\text{Dropout}(\boldsymbol{\theta})}(\boldsymbol{x})\right)\right), \tag{9}$$

where $\lambda$ controls the augmentation strength. We note that $\boldsymbol{x}$ in Equation 9 can be either prototypes from old classes ($\boldsymbol{x}^{\text{old}}$) or samples from new classes ($\boldsymbol{x}^{\text{new}}$). We term the former case as DOA-old and the later one as DOA-new. For DOA-old, we perform the inference twice for prototypes of old classes, with one full

inference to output $h_{\boldsymbol{\theta}}(\boldsymbol{x}^{\text{old}})$ and one Dropout inference (where neurons are randomly masked to zero) to output $h_{\text{Dropout}(\boldsymbol{\theta})}(\boldsymbol{x})$. The intuition behind DOA-new is to take advantage of the representation richness in the new classes and transfer such information to diversify the representation space of old classes. Therefore, DOA-new takes both the model $\boldsymbol{\theta}$ and the feature manifold of the new classes into account. By considering the feature manifold of the new classes, we expect that DOA-new can introduce more informative gradients that aims to separate features from old classes and features from new classes.

**Weight-adversarial perturbation.** While Dropout randomly perturbs the network parameters $\boldsymbol{\theta}$, we propose an alternative way to perturb $\boldsymbol{\theta}$ – adversarially generate $\Delta\boldsymbol{\theta}$ in the first perturbation model (Equation 8). Specifically, we have that $\tilde{\Delta}\boldsymbol{f} = \lambda_1 h_{\boldsymbol{\theta}}(\boldsymbol{x}) - \lambda_2 h_{\boldsymbol{\theta}+\Delta\boldsymbol{\theta}}(\boldsymbol{x})$ where $\Delta\boldsymbol{\theta}$ is generated adversarially to minimize the confidence of the new class. Then we use the following model in WAP:

$$\boldsymbol{f} = \left\| h_{\boldsymbol{\theta}}(\boldsymbol{x}^{\text{old}}) \right\| \cdot \mathcal{P}_{\mathbb{S}} \left( \mathcal{P}_{\mathbb{S}} \left( h_{\boldsymbol{\theta}}(\boldsymbol{x}^{\text{old}}) \right) + \lambda \cdot \mathcal{P}_{\mathbb{S}} \left( h_{\boldsymbol{\theta}+\Delta\boldsymbol{\theta}}(\boldsymbol{x}^{\text{old}}) \right) \right), \tag{10}$$

where we obtain $\Delta\boldsymbol{\theta}$ by performing projected gradient descent to optimize the following objective:

$$\Delta\boldsymbol{\theta} = \arg\min_{\|\Delta\boldsymbol{\theta}\|\leq\epsilon} \mathcal{L}_{\text{ce}} \left( g_{\boldsymbol{\phi}} \left( h_{\boldsymbol{\theta}+\Delta\boldsymbol{\theta}}(\boldsymbol{x}^{\text{old}}) \right), y^{\text{new}} \right), \tag{11}$$

where $\Delta\boldsymbol{\theta}$ is defined as a solution. In general, WAP aims to perturb $\boldsymbol{\theta}$ by generating $\Delta\boldsymbol{\theta}$ in an adversarial fashion such that the input data can move closer to the decision boundary.

Despite focusing on different applications, WAP has intrinsic connections to (Wu et al., 2020; Zheng et al., 2021) where adversarial weight perturbation is shown to converge to flat minima (Hochreiter & Schmidhuber, 1997) and is beneficial to generalization. Different from (Wu et al., 2020; Zheng et al., 2021), WAP perturbs the network weights adversarially based on the new class label $y^{\text{new}}$, making it good at preventing catastrophic forgetting. WAP first finds a new set of network weights $\boldsymbol{\theta}_{\text{adv}} = \boldsymbol{\theta} + \Delta\boldsymbol{\theta}$ that catastrophically forget the knowledge of old classes and confuse the old classes with the new ones. Then WAP uses $\boldsymbol{\theta}_{\text{adv}}$ to generate intra-class perturbation such that the model can be regularized away from $\boldsymbol{\theta}_{\text{adv}}$ and the concepts of old and new classes can be better separated. In general, WAP seeks to model the intra-class variation following the direction to the decision boundary (as opposed to the isotropic direction in DOA), which leads to more informative gradients to learn discriminative features that well separate old and new classes.

**Variation transfer.** We now consider the second perturbation model in Equation 8. Our core idea is to transfer the variation in new classes to diversify the intra-class variation in the old classes. Specifically in Equation 8(II), we let $\boldsymbol{x}$ be a virtual sample that corresponds to the mean feature of the new class (*i.e.*, $\tilde{\boldsymbol{x}}^{\text{new}}$) and $\Delta\boldsymbol{x}$ be the difference between individual feature in the new class and the virtual sample (*i.e.*, $\boldsymbol{x}^{\text{new}} - \tilde{\boldsymbol{x}}^{\text{new}}$). After simplifying hyperparameters, we have the following model for VT:

$$\boldsymbol{f} = \left\| h_{\boldsymbol{\theta}}(\boldsymbol{x}^{\text{old}}) \right\| \cdot \mathcal{P}_{\mathbb{S}} \left( \mathcal{P}_{\mathbb{S}} \left( h_{\boldsymbol{\theta}}(\boldsymbol{x}^{\text{old}}) \right) + \lambda \cdot \mathcal{P}_{\mathbb{S}} \left( (h_{\boldsymbol{\theta}}(\boldsymbol{x}^{\text{new}}) - h_{\boldsymbol{\theta}}(\tilde{\boldsymbol{x}}^{\text{new}})) \right) \right), \tag{12}$$

where $h_{\boldsymbol{\theta}}(\tilde{\boldsymbol{x}}^{\text{new}})$ denotes the mean feature that is also equal to $\bar{h}_{\boldsymbol{\theta}}(\boldsymbol{x}^{\text{new}}) = \mathbb{E}_{\boldsymbol{x}^{\text{new}}} h_{\boldsymbol{\theta}}(\boldsymbol{x}^{\text{new}})$. VT implicitly imposes an assumption that intra-class representation variations for different classes are similar.

### 4.4 Discussions and Intriguing Insights

**Connection to large-margin softmax.** MOCA demonstrates the effectiveness of modeling intra-class representation in continual learning. The intuition of why MOCA can well regularize the representation space also comes from the series of works in large-margin softmax (Liu et al., 2016; 2017a;b; 2022; Wang et al., 2018a;b; Deng et al., 2019). The central idea of large-margin softmax can be interpreted as constructing a hard virtual feature that is close to the decision boundary, and optimizing this virtual sample amounts to creating large between-class margins (see justification in Appendix C). MOCA adopts a similar approach to create margins between old classes and new classes such that the old classes will not be forgotten catastrophically.

**Why intra-class modeling is difficult yet possible?** Modeling intra-class representation is in general a highly nontrivial task, because it requires us to jointly consider input distribution, properties of neural networks, objective functions and optimizers. Fortunately, recent theoretical studies (*e.g.*, neural collapse (Papyan et al., 2020; Lu & Steinerberger, 2022) and uniformity (Wang & Isola, 2020; Liu et al., 2021c)) discover a regularity

of hyperspherical uniformity in the representation space. Moreover, empirical studies (Liu et al., 2018b; Chen et al., 2020) also show that the intra-class representation is distributed like a vMF distribution. Motivated by these studies, MOCA proposes a unified framework and specific algorithms to model intra-class variation.

**Open problems**. We only consider a few simple variants and it remains an open problem to design a better variant. Another open problem is the structure of representation space. Stronger regularities for intra-class modeling (*e.g.*, discrete (Van Den Oord et al., 2017), causal Schölkopf et al. (2021)) may improve MOCA.

## 5  Experiments and Results

**Experimental settings.**   In this section, we evaluate existing competitive baselines and different MOCA variants on CIFAR-10 (Krizhevsky et al., 2009), CIFAR-100 (Krizhevsky et al., 2009) and TinyImageNet (Deng et al., 2009). We consider three different continual learning settings, *i.e.*, (1) offline continual learning, (2) online continual learning, and (3) proxy-based continual learning. For offline and online continual learning, we divide the dataset into five tasks for CIFAR-10 (two classes per task) and CIFAR-100 (20 classes per task), and divide the dataset into 10 tasks for TinyImageNet (20 classes per task). For the proxy-based continual learning, we follow the same experimental setting as in (Zhu et al., 2021). To facilitate the encoder learning, we further perform hyperspherical projection to all the learned features $f$ for all the compared methods (*i.e.*, make linear classifiers $g(\cdot)$ fully rely on angles), following (Wang et al., 2018a). Full experimental and implementation details can be found in Appendix A. Additional experiments are given in Appendix D.

### 5.1  Empirical Comparison of Different MOCA Variants

We evaluate different variants of MOCA based on a simple and clean baseline – ER (Riemer et al., 2018). We show in Figure 1 that all the MOCA variants can effectively increase the intra-class variation for old classes. Table 1 shows that most of MOCA variants can consistently improve continual learning. While both model-agnostic and model-based MOCA can

| Setting | Baseline | Gaussian | vMF | DOA-old | DOA-new | VT | WAP |
|---|---|---|---|---|---|---|---|
| Offline | 31.08 | 37.29 | **38.76** | 33.67 | 38.75 | 39.78 | **41.02** |
| Online | 31.90 | **32.78** | 31.25 | 30.20 | 29.48 | 32.55 | **33.72** |
| Proxy | 31.26 | **42.54** | 42.24 | - | 45.72 | **46.77** | - |

Table 1: Comparison of different MOCA variants in 3 continual settings on CIFAR-100. Classification accuracy (%) on the full testing set is reported. Results are averaged with 3 random seeds and the best ones are marked in bold.

achieve significantly better accuracy than the baseline, we observe that model-based MOCA generally outperforms both model-agnostic MOCA and the baseline by a considerable margin. For the family of model-agnostic MOCA, Gaussian and vMF achieve similar performance, but sampling Gaussian distribution yields better efficiency and simplicity. For the family of model-based MOCA, WAP achieves the best performance in both offline and online continual learning, while VT performs the best in proxy-based continual learning. The proxy-based continual learning setting does not allow the memory replay, so both DOA-old and WAP can no longer be applied. Although all the MOCA variants can increase intra-class variation, we note that it does not necessarily lead to better performance. Therefore, the specific direction to model the intra-class variation is also important, which causes the performance difference for DOA, VT and WAP.

### 5.2  How Perturbation Magnitude Affects Performance

We evaluate how perturbation magnitude influences different MOCA variants on CIFAR-100. The experimental settings are the same as Section 5.1 (the offline setting). Results in Figure 8 show that most of the MOCA variants achieve consistent improvement under a wide range of different perturbation magnitudes. Specifically, we consider two ways to measure the perturbation magnitude: (1) the hyperparameter $\lambda$ in Section 4.3 and (2) the hard constraint that the perturbation has a fixed angle to the pro-

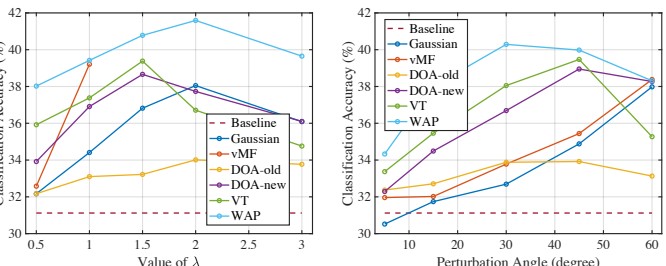

Figure 8: Left: the hyperparameter $\lambda$ vs. classification accuracy. Right: the perturbation angle vs. classification accuracy.

totype feature (this is achieved by performing simple spherical projection after applying the specific MOCA variant). We observe that model-based MOCA performs generally better than model-agnostic MOCA under most perturbation magnitudes, and WAP is the best-performing variant under all magnitude constraints.

## 5.3 MOCA Learns Discriminative Classifiers

One of the most significant problems in memory-based continual learning is the classifier bias caused by the highly imbalanced dataset (and imbalanced mini-batches in training). MOCA addresses this by diversifying the representation space of old classes. In order to visually compare MOCA and the baseline, we compute the average pair-wise angle between learned classifier vectors in two tasks. For example,

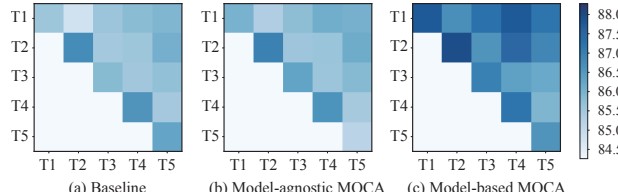

Figure 9: Average angle between the classifiers. Ti denotes the $i$-th task. The color of blocks shows the average classifier angle between different tasks. Only the upper triangular part is shown.

the block at the T2 column and the T1 row shows the average pair-wise angle between classifiers in the first task and classifiers in the second task. Figure 9 gives the results. We note that a large angle between classifiers not only indicates more discriminativeness in classifiers themselves, but it also implies inter-class separability in the representation space because more separable features generally lead to more separable classifiers Liu et al. (2022). The results show that both model-agnostic MOCA (*i.e.*, Gaussian) and model-based MOCA (*i.e.*, WAP) can effectively improve the discriminativeness of the learned classifiers in continual learning.

## 5.4 Comparison to State-of-the-art Methods

We conduct a comprehensive comparison of existing state-of-the-art methods in three settings including offline, online, and proxy-based continual learning. We use CIFAR-10, CIFAR-100, and TinyImageNet as the benchmark datasets.

**Offline continual learning.** We apply the best-performing variant of model-agnostic MOCA (*i.e.*, Gaussian) and model-based MOCA (*i.e.*, WAP) to three baselines (*i.e.*, ER (Riemer et al., 2018), DER++ (Buzzega et al., 2020), ER-ACE (Caccia et al., 2021)). Table 2 shows the results with three buffer sizes (200, 500 and 2000) on CIFAR-10, CIFAR-100 and TinyImageNet. The results consistently validate the effectiveness of both model-agnostic and model-based MOCA. On CIFAR-10, we observe that Gaussian greatly improves ER while only achieving incremental improvement on both DER++ and ER-ACE. This further emphasizes the importance of perturbation directions rather the absolute variance. In comparison, WAP improves all three baselines (ER, DER++ and ER-ACE) by a large margin (∼10% in some cases). On both CIFAR-100 and TinyImageNet, WAP still consistently improves all three baselines by a considerable margin. Gaussian is able to improve the performance of ER and ER-ACE, while only being comparable (sometimes even worse) in the case of DER++. We suspect that the distillation step in DER++ is not suitable for Gaussian perturbation, and to amend this, we may need to store all the logits for the perturbed features, which is memory-expensive. In contrast, WAP can work well with DER++, because it only perturbs along the most informative and boundary-dependent directions. Moreover, the improvement of MOCA is consistent on all three sizes of memory buffers. Applying

| CIFAR-10 | | | |
|---|---|---|---|
| Method | $M$=200 | $M$=500 | $M$=2000 |
| GEM (Lopez-Paz & Ranzato, 2017) | 29.99±3.92 | 29.45±5.64 | 27.20±4.50 |
| GSS (Aljundi et al., 2019b) | 38.62±3.59 | 48.97±3.25 | 60.40±4.92 |
| iCaRL (Rebuffi et al., 2017) | 32.44±0.93 | 34.95±1.23 | 33.57±1.65 |
| ER (Riemer et al., 2018) | 49.07±1.65 | 61.58±1.12 | 76.89±0.99 |
| **ER w/ Gaussian** | 61.52±1.42 | 68.54±2.01 | 78.27±0.52 |
| **ER w/ WAP** | **63.12±2.15** | **72.07±1.37** | **80.38±0.95** |
| DER++ (Buzzega et al., 2020) | 64.88±1.17 | 72.70±1.36 | 78.54±0.97 |
| **DER++ w/ Gaussian** | 63.02±0.53 | 71.04±0.72 | 79.22±0.42 |
| **DER++ w/ WAP** | **65.12±0.77** | **75.01±0.24** | **81.54±0.12** |
| ER-ACE (Caccia et al., 2021) | 63.18±0.56 | 71.98±1.30 | 80.01±0.76 |
| **ER-ACE w/ Gaussian** | 65.21±0.89 | 72.01±0.76 | 78.92±0.58 |
| **ER-ACE w/ WAP** | **66.56±0.81** | **72.86±1.02** | **80.24±0.50** |

| CIFAR-100 | | | |
|---|---|---|---|
| Method | $M$=200 | $M$=500 | $M$=2000 |
| GEM (Lopez-Paz & Ranzato, 2017) | 20.75±0.66 | 25.54±0.65 | 37.56±0.87 |
| GSS (Aljundi et al., 2019b) | 19.42±0.29 | 21.92±0.34 | 27.07±0.25 |
| iCaRL (Rebuffi et al., 2017) | 28.00±0.91 | 33.25±1.25 | 42.19±2.42 |
| ER (Riemer et al., 2018) | 22.14±0.42 | 31.02±0.79 | 43.54±0.59 |
| **ER w/ Gaussian** | 27.51±0.93 | 37.54±0.71 | 49.61±1.01 |
| **ER w/ WAP** | **30.16±1.02** | **40.24±0.78** | **52.92±0.03** |
| DER++ (Buzzega et al., 2020) | 29.68±1.38 | 39.08±1.76 | 54.38±0.86 |
| **DER++ w/ Gaussian** | 30.59±0.40 | 40.52±0.29 | 53.7±0.42 |
| **DER++ w/ WAP** | **32.18±0.67** | **43.78±0.89** | **55.04±0.81** |
| ER-ACE (Caccia et al., 2021) | 35.09±0.92 | 43.12±0.85 | 53.88±0.42 |
| **ER-ACE w/ Gaussian** | 37.01±0.70 | 44.57±0.83 | 54.84±0.12 |
| **ER-ACE w/ WAP** | **37.46±0.77** | **45.79±0.73** | **56.02±0.64** |

| TinyImageNet | | | |
|---|---|---|---|
| Method | $M$=200 | $M$=500 | $M$=2000 |
| GEM (Lopez-Paz & Ranzato, 2017) | - | - | - |
| GSS (Aljundi et al., 2019b) | 8.57±0.13 | 9.63±0.14 | 11.94±0.17 |
| iCaRL (Rebuffi et al., 2017) | 5.50±0.52 | 11.00±0.55 | 18.10±1.13 |
| ER (Riemer et al., 2018) | 8.65±0.16 | 10.05±0.28 | 18.19±0.47 |
| **ER w/ Gaussian** | 9.42±0.12 | 12.94±0.52 | 21.43±0.78 |
| **ER w/ WAP** | **10.41±0.37** | **16.27±0.25** | **22.62±0.10** |
| DER++ (Buzzega et al., 2020) | 10.96±1.17 | 19.38±1.41 | **30.11±0.57** |
| **DER++ w/ Gaussian** | 10.52±0.12 | 15.75±0.35 | 25.28±0.30 |
| **DER++ w/ WAP** | **12.07±0.35** | **21.24±0.47** | 29.33±0.71 |
| ER-ACE (Caccia et al., 2021) | 14.29±0.74 | 20.87±0.69 | 30.10±0.92 |
| **ER-ACE w/ Gaussian** | 16.72±0.41 | 22.82±0.39 | 30.92±0.41 |
| **ER-ACE w/ WAP** | **17.05±0.22** | **23.56±0.85** | **32.54±0.72** |

Table 2: Offline continual learning on CIFAR-10, CIFAR-100 and TinyImageNet. Final classification accuracy (%) on the full testing set is given. Results are averaged with 3 random seeds.

MOCA to the simplest baseline (ER) already leads to comparable performance to the state-of-the-art methods.

**Online continual learning.** We apply the best-performing MOCA variant (*i.e.*, WAP) to online continual learning. The results are shown in Table 3. In the online setting, all the previously seen data from this or previous tasks are not accessible. We evaluate WAP using two different buffer sizes (20 and 100) on CIFAR-10, CIFAR-100 and MiniImageNet. We apply WAP on three baselines: ER, DER++ and ER-ACE. We draw a few conclusions from the results: (1) WAP can consistently improve all three baselines by a considerable margin in most of the settings. This again validates the effectiveness of MOCA.

| Method | CIFAR-10 | | CIFAR-100 | | MiniImageNet | |
|---|---|---|---|---|---|---|
| | $M$=20 | $M$=100 | $M$=20 | $M$=100 | $M$=20 | $M$=100 |
| A-GEM (Chaudhry et al., 2018) | 18.56 | 18.60 | 3.50 | 3.26 | 2.94 | 3.04 |
| MIR (Aljundi et al., 2019a) | 24.20 | 45.44 | 12.70 | 17.50 | 11.54 | 11.48 |
| SS-IL (Ahn et al., 2021) | 35.54 | 42.78 | 16.20 | 26.24 | 16.96 | 24.38 |
| iCaRL (Rebuffi et al., 2017) | 40.94 | 49.76 | 17.55 | 19.86 | 12.30 | 15.20 |
| ER (Riemer et al., 2018) | 31.90 | **42.48** | 13.52 | **25.24** | 16.16 | **21.62** |
| **ER w/ WAP** | **33.72** | 41.26 | **13.90** | 23.29 | **17.22** | 19.52 |
| DER++ (Buzzega et al., 2020) | 34.36 | 43.38 | 12.84 | 13.74 | **17.00** | 18.56 |
| **DER++ w/ WAP** | **41.56** | **45.64** | **18.76** | **22.54** | 16.32 | **19.20** |
| ER-ACE (Caccia et al., 2021) | 42.90 | 53.88 | 16.88 | 27.48 | 21.00 | 28.96 |
| **ER-ACE w/ WAP** | **43.57** | **54.42** | **18.90** | **29.52** | **22.56** | **29.70** |

Table 3: Online continual learning on CIFAR-10, CIFAR-100 and TinyImageNet. Final classification accuracy (%) on the full testing set is given. Results are averaged with 3 random seeds.

(2) In comparison, the performance gain of WAP in the online setting is less significant than that in the offline setting. The reason behind this is that online continual learning suffers not only from catastrophic forgetting but also from the under-fitting of the incoming data which can only be seen once, while MOCA can only help with catastrophic forgetting rather than data under-fitting.

**Proxy-based continual learning.** Since the memory buffer is disabled in proxy-based continual learning (Zhu et al., 2021), WAP cannot be used. However, we can still explore the effectiveness of modeling intra-class variation in this scenario with the other MOCA variants (*e.g.*, DOA and VT). In the proxy-based continual learning, PASS (Zhu et al., 2021) proposed to augment the old-class prototype by adding Gaussian

| Method | CIFAR-100 | | | MiniImageNet | | |
|---|---|---|---|---|---|---|
| | $T$=5 | $T$=10 | $T$=20 | $T$=5 | $T$=10 | $T$=20 |
| PASS (Zhu et al., 2021) | 64.39 | 57.36 | 58.09 | 48.26 | 46.54 | 42.09 |
| **PASS w/ DOA** | 66.82 | 63.30 | 62.62 | 47.92 | 47.55 | 47.11 |
| **PASS w/ VT** | **67.75** | **63.64** | **63.09** | **48.35** | **47.90** | **47.33** |

Table 4: Proxy-based continual learning on CIFAR-100 and MiniImageNet. Final accuracy (%) on the full testing set is given. Results are averaged with 3 random seeds.

noise, which is conceptually similar to our Gaussian MOCA. Therefore, PASS can be viewed as a special case of Gaussian MOCA. Table 4 shows that both DOA and VT perform better than PASS, further validating our conclusion that model-based MOCA works better than model-agnostic MOCA.

**Performance over continual tasks**. We also plot the average testing accuracy of currently seen tasks when learning different continual tasks in the offline setting. We perform the experiments on CIFAR-100 (5 tasks with 20 classes per task) and TinyImageNet (10 tasks with 20 classes per task) with buffer size 500. From Figure 10, we can observe that all the MOCA variants perform better than the ER baseline by a considerable margin and WAP

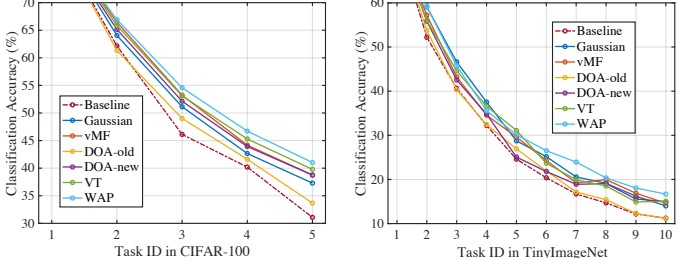

Figure 10: Average testing accuracy (%) over continual tasks.

again works the best across different training phases. While achieving better performance than baseline, DOA-old does not perform as well as DOA-new. We suspect that the variation created by Dropout on the old data is less diverse and less informative than that on the new data.

## 5.5 Comparison to Gradient and Representation Diversification

Besides methods in continual learning, we also compare MOCA to some popular methods that can diversify gradients and representations. Specifically, we consider loss balancing methods such as Re-weighting (Kang et al., 2019) and Focal Loss (Lin et al., 2017), and representation augmentation methods such as Manifold

| Method | CIFAR-10 | | | CIFAR-100 | | |
|---|---|---|---|---|---|---|
| | $k$=200 | $k$=500 | $k$=2000 | $k$=200 | $k$=500 | $k$=2000 |
| ER (Riemer et al., 2018) | 49.07 | 61.58 | 76.89 | 21.71 | 28.12 | 43.10 |
| + Re-weighting (Kang et al., 2019) | 53.02 | 66.54 | 77.92 | 24.58 | 30.12 | 44.31 |
| + Focal Loss (Lin et al., 2017) | 46.07 | 60.97 | 77.26 | 22.43 | 27.19 | 43.37 |
| + Manifold Mixup (Verma et al., 2019) | 55.21 | 67.02 | 77.54 | 23.97 | 29.33 | 45.21 |
| **+ Model-based MOCA (WAP)** | **63.12** | **72.07** | **80.38** | **30.16** | **40.24** | **52.92** |

Table 5: Comparisons of the proposed approaches with existing classical loss re-weighting methods. The best results are marked in bold.

Mixup (Verma et al., 2019). We apply these methods (with the best-performing hyperparameters) to ER and compare them with our best-performing MOCA variant (WAP). Results in Table 5 show that WAP can outperform all these methods on both CIFAR-10 and CIFAR-100 under three different buffer sizes (200, 500 and 2000). The experiments also support our argument that gradient diversity for old classes is of great importance to continual learning.

## 6 Concluding Remarks

In this work, we study the problem of memory-based continual learning. Due to few old-class data samples, we observe that there exists a serious lack of diversity in the representation space for old classes. This behavior causes *representation collapse* for old classes and an intra-class variation gap between old classes and new classes. This representation collapse further causes *gradient collapse* which prevents the model to acquire effective information for remembering old classes and leads to catastrophic forgetting. To address this, we propose the MOCA framework to model the intra-class variation and improve continual learning. We propose several variants of model-agnostic MOCA and model-based MOCA under this framework. In all continual learning settings, we show that all the MOCA variants can serve as a plug-and-play component to effortlessly improve a number of existing continual learning methods, demonstrating the effectiveness of MOCA.

## Acknowledgements

[*]WL and LH share the corresponding authorship. WL acknowledges support from the Cambridge-Tübingen PhD fellowship, the German Federal Ministry of Education and Research (BMBF): Tübingen AI Center, FKZ: 01IS18039A, 01IS18039B; and by the Machine Learning Cluster of Excellence, EXC number 2064/1 – Project number 390727645. AW acknowledges support from a Turing AI Fellowship under EPSRC grant EP/V025279/1, The Alan Turing Institute, and the Leverhulme Trust via CFI. We gratefully acknowledge the support of MindSpore, CANN (Compute Architecture for Neural Networks) and Ascend AI Processor used for this research.

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

# Appendix

## Table of Contents

# A   Implementation Details

**Compared Baseline.** For offline continual learning, we include several classical approaches: GEM (Lopez-Paz & Ranzato, 2017) is the gradient projection method; GSS (Aljundi et al., 2019b) considers the variance of the memory buffer; and iCARL (Rebuffi et al., 2017) uses the previous model for distillation regularization. For online continual learning, the compared approaches include: A-GEM (Chaudhry et al., 2018), an online version of GEM, MIR (Aljundi et al., 2019a), a buffer selection method considering the sample influence to performance and SS-IL (Ahn et al., 2021), a method separating the old-class and new-class in softmax are considered. For both the offline and online settings, we plug our method into several classical and comparative methods, ER (Riemer et al., 2018), randomly selecting buffer for replay, DER++ (Buzzega et al., 2020), using the dark knowledge of previous logits for a better distillation regularization, ER-ACE (Caccia et al., 2021), a new method preventing the overwhelming negative gradient for old-class. For proxy-based continual learning, we follow PASS (Zhu et al., 2021) and add our proposed approaches on PASS.

**Evaluation Metrics and proposed approaches.** We use the ResNet18 (He et al., 2016) as the backbone and use the test classification accuracy of the final continual task as the metric. Although all the various approaches we proposed, Gaussian, DOA-old, DOA-new, VT, and WAP have achieved considerable performance gains, we use the WAP as our final approach to compare with existing methods. The detailed implementation of WAP shows in Algorithm 1. In our experiments, we set the inner learning rate $\zeta$ as 10 and the inner iteration number $T$ as 1.

**Offline continual learning.** For offline continual learning, we implement our MOCA based on the code of DER (Buzzega et al., 2020). For all the MOCA approaches, the perturbation magnitude $\lambda$ is set as 2.0, which shows the best empirical performance. For DOA-old and DOA-new, the dropout rate is set as 0.5. For WAP, we update the proxy model $\boldsymbol{\theta}_p$ in each iteration, and the proxy loss weight is set as 10. After producing the perturbed feature, the proxy model will be reloaded as the original model. For both the compared baselines and proposed approaches, the training epoch is set as 50. The batch size is set as 32 and the initial learning rate is est as 0.1. All the other settings are the same as the DER (Buzzega et al., 2020).

**Online continual learning.** For proxy-based continual learning, we implement our MOCA based on the code of ER-ACE (Caccia et al., 2021). Since learning efficiency is also important for online continual learning, for all the MOCA approaches, the perturbation magnitude $\lambda$ is set as 0.8. The batch size is set as 10. The initial learning rate is est as 0.1. All the other settings are the same as the ER-ACE (Caccia et al., 2021).

**Proxy-based continual learning.** For proxy-based continual learning, we implement our MOCA based on the code of PASS (Zhu et al., 2021). The perturbation magnitude $\lambda$ is set as 1.0. The other hyper-parameter and continual learning settings follow PASS.

For the complete implementation details and settings, please refer to our official PyTorch implementation at https://github.com/yulonghui/MOCA.

---

**Algorithm 1** Weight-Adversarial Perturbation

---

**Require:** New task training set $\mathcal{D} = \{(\boldsymbol{x}^{new}, \boldsymbol{y}^{new})\}$, New data batch size $n$, Old task buffer set $\mathcal{M} = \{(\boldsymbol{x}^{old}, \boldsymbol{y}^{old})\}$, Old data batch size $m$, Loss function $\ell$, Initial model parameter $\boldsymbol{\theta}_0$, Initial proxy model parameter $\boldsymbol{\theta}_{adv}$, Outer learning rate $\eta$, Inner learning rate $\zeta$, Inner iteration number $T$, L$_2$ norm ball radius $\epsilon$

1: **while** $\boldsymbol{\theta}_k$ not converged **do**
2:     Update iteration: $k \leftarrow k + 1$
3:     Sample $\mathcal{B}_m = \{(\boldsymbol{x}_i^{old}, \boldsymbol{y}_i^{old})\}_{i=1}^m$ from buffer set $\mathcal{M}$
4:     Initialize proxy model: $\boldsymbol{\theta}_{adv} \leftarrow \boldsymbol{\theta}_k$
5:     Initialize perturbation: $\Delta_{\mathcal{B}_m} \leftarrow \mathbf{0}$
6:     **for** $t \leftarrow 1$ to $T$ **do**
7:         Select $m$ random new labels: $\boldsymbol{y^{adv}} = \{\boldsymbol{y}_i^{new}\}_{i=1}^m$
8:         Compute gradient:
            $\nabla \mathcal{J}_{\text{ADV}, \mathcal{B}_m} \leftarrow \sum_{i=1}^m \nabla_{\boldsymbol{\theta}_{adv}} \ell(\boldsymbol{x}_i^{old}, \boldsymbol{y}_i^{adv}; \boldsymbol{\theta}_{adv} + \Delta_{\mathcal{B}_m})/m$
9:         Update perturbation: $\Delta_{\mathcal{B}_m} \leftarrow \Delta_{\mathcal{B}_m} - \zeta \nabla \mathcal{J}_{\text{ADV}, \mathcal{B}_m}$
10:        **if** $\|\Delta_{\mathcal{B}_m}\|_2 > \epsilon$ **then**
11:            Normalize perturbation: $\Delta_{\mathcal{B}_m} \leftarrow \epsilon \Delta_{\mathcal{B}_m} / \|\Delta_{\mathcal{B}_m}\|_2$
12:        **end if**
13:    **end for**
14:    Update proxy model: $\boldsymbol{\theta}_{adv} \leftarrow \boldsymbol{\theta}_{adv} + \Delta_{\mathcal{B}_m}$
15:    Compute gradient:
        $\boldsymbol{f} = \|h_{\boldsymbol{\theta}_k}(\boldsymbol{x}^{old})\| \cdot \mathcal{P}_{\mathbb{S}} \left( \mathcal{P}_{\mathbb{S}} \left( h_{\boldsymbol{\theta}_k}(\boldsymbol{x}^{old}) \right) + \lambda \cdot \mathcal{P}_{\mathbb{S}} \left( h_{\boldsymbol{\theta}_{adv}}(\boldsymbol{x}^{old}) \right) \right)$
        $\nabla \mathcal{J}_{\text{WAP}, \mathcal{B}_m} \leftarrow \sum_{i=1}^m \nabla_{\boldsymbol{\theta}_k} \ell(g_{\boldsymbol{\phi}_k}(\boldsymbol{f}_i), \boldsymbol{y}_i^{old}; \boldsymbol{\theta}_k)/m$
16:    Sample $\mathcal{B}_n = \{(\boldsymbol{x}_i^{new}, \boldsymbol{y}_i^{new})\}_{i=1}^n$ from training set $\mathcal{D}$
17:    Compute gradient:
        $\nabla \mathcal{J}_{\text{WAP}, \mathcal{B}_n} \leftarrow \sum_{i=1}^n \nabla_{\boldsymbol{\theta}_k} \ell(\boldsymbol{x}_i^{new}, \boldsymbol{y}_i^{new}; \boldsymbol{\theta}_k)/n$
18:    Update parameter: $\boldsymbol{\theta}_{k+1} \leftarrow \boldsymbol{\theta}_k - \eta(\nabla \mathcal{J}_{\text{WAP}, \mathcal{B}_m} + \nabla \mathcal{J}_{\text{WAP}, \mathcal{B}_n})$
19: **end while**

---

# B MOCA as Implicit Data Augmentation

As illustrated in Section 3.3, MOCA can be seen as a kind of implicit data augmentation. ISDA (Wang et al., 2019) has proposed a framework to show the semantic changes and map the features back to the pixel space. The visualization framework proposed in ISDA is shown in Figure 11. In the first step, we load the pre-trained BigGAN (Brock et al., 2018) for the generator and fixed. Then we minimize the discrepancy between real image, $\boldsymbol{x}$ and generated image $g(\boldsymbol{z})$ to optimize the prior distribution $\boldsymbol{z}$. In the second step, we force the feature of generated image $h(g(\boldsymbol{z}))$ and the augmented feature $\boldsymbol{f}$ to be close. By doing this, we can further optimize $\boldsymbol{z}$ and show fake image $g(\boldsymbol{z})$ corresponding to the augmented feature explicitly. More details can be found in ISDA (Wang et al., 2019).

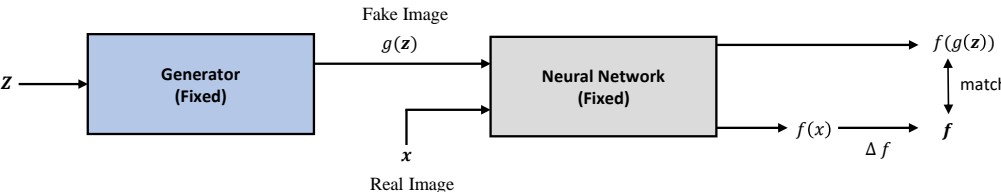

Figure 11: The visualization framework used in MOCA.

## C  Connection to Large-margin Softmax

We denote the feature as $\boldsymbol{x}_i$, its label as $y_i$, the $i$-th classifier as $\boldsymbol{w}_i$, and the angle between $\boldsymbol{x}_i$ and $\boldsymbol{w}_j$ as $\theta_j$. Then we can write the large-margin cross-entropy loss (Liu et al., 2016) as

$$\mathcal{L}_{\text{Large-Margin}} = \sum_i \frac{\exp\left(\|\boldsymbol{w}_{y_i}\| \cdot \|\boldsymbol{x}_i\| \cdot \cos(\theta_{y_i} + \Delta\theta)\right)}{\sum_j \exp\left(\|\boldsymbol{w}_j\| \cdot \|\boldsymbol{x}_i\| \cdot \cos(\theta_j + \mathbf{1}(j = y_i) \cdot \Delta\theta)\right)} \tag{13}$$

where we have that $\Delta\theta = (m-1)\theta$. There are many other possible forms for $\Delta\theta$ in practice (Liu et al., 2022). For the cross-entropy loss with MOCA, we have the following form:

$$\mathcal{L}_{\text{MOCA}} = \sum_i \frac{\exp\left(\|\boldsymbol{w}_{y_i}\| \cdot \|\boldsymbol{x}_i\| \cdot \cos(\theta_{y_i} + \Delta\theta_{y_i})\right)}{\sum_j \exp\left(\|\boldsymbol{w}_j\| \cdot \|\boldsymbol{x}_i\| \cdot \cos(\theta_j + \Delta\theta_j)\right)} \tag{14}$$

where adding perturbation to the feature $\boldsymbol{x}$ results in a series of angular deviations $\Delta\theta_j, \forall j$. From the two loss formulations above, we can see that the difference mostly lies in the $\Delta\theta_j, \forall j \neq y_i$. For the large-margin loss, we have that $\Delta\theta_j = 0, \forall j \neq y_i$. Let $\Delta\theta = \Delta\theta_{y_i}$, and we will easily have that (assuming all perturbed angles are within $[0, \pi]$ and $\Delta\theta_j \geq 0, \forall j$)

$$\mathcal{L}_{\text{Large-Margin}} \leq \mathcal{L}_{\text{MOCA}}. \tag{15}$$

If $\Delta\theta_j \leq 0, \forall j$, we have that

$$\mathcal{L}_{\text{Large-Margin}} \geq \mathcal{L}_{\text{MOCA}}. \tag{16}$$

The perturbation in MOCA happens in $\Delta\theta_{y_i}$ and the rest $\Delta\theta_j, \forall j \neq y_i$ are simply the consequence of this perturbation. Therefore, $|\Delta\theta_{y_i}| \geq |\Delta\theta_j|, \forall j \neq y_i$ always holds. Then it can be approximately viewed that $\mathcal{L}_{\text{Large-Margin}} \approx \mathcal{L}_{\text{MOCA}}$ under some cases.

# D  Additional Experimental Results and Discussions

## D.1  MOCA Diversifies the Collapsed Gradient

In this work, as expounded in the section 1, a serious problem in continual learning causing catastrophic forgetting is that the training gradients are not diversified and collapse in some directions and this lack-of-diversity problem causes poor performance. As shown in Figure 12, all of our proposed approaches can diversify the gradient direction to the same extent. DOA-new is better than DOA-old in terms of improving gradient diversity. VT, DOA-new, and WAP all consider the new-class information and have a better gradient diversity approximation to the joint training method, which also validates our motivation – modeling the intra-class variation by the model-conditioned data manifold and considering the new-class information can better resist forgetting and approximate the gradient to the joint training.

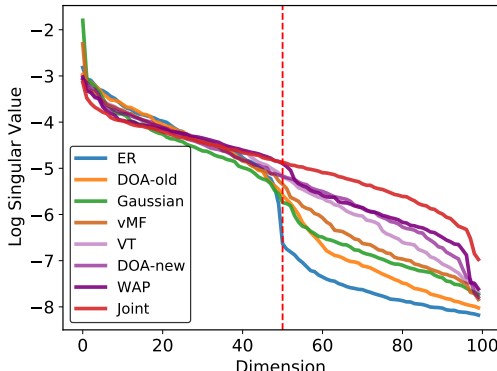

Figure 12: The singular value of the training gradients for different methods. We show the 2-continual learning task for CIFAR-100, where 50 classes are old-classes, and the other 50 classes are new-classes. All the VT, DOA-new, and WAP consider the new-class information.

## D.2  Variation Towards New-Class is Important for Continual Learning

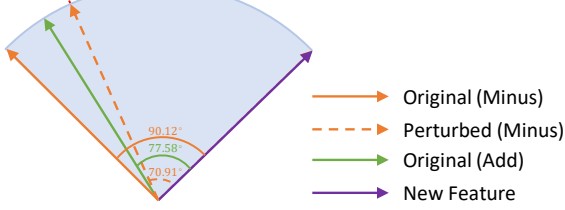

| Method | Perturbed | Original | Accuracy |
|---|---|---|---|
| Baseline | - | 72.51 | 29.94 |
| Minus New Feature | 90.12 | 70.91 | 27.35 |
| Add New Feature | 71.34 | **77.58** | **32.60** |

Table 6: Adding perturbations in different directions: Towards the new-class feature or opposite to the new-class feature.

Figure 13: Different changes of the angle between old-class and new-class features by diversifying the feature towards or opposite the new-class manifold.

Although an Isotropic Gaussian Noise can help improve the variance of representation space of old-class and the empirical performance, the direction of perturbation is also the key to resisting forgetting. We naively add or minus the feature of new classes to that of old classes, which means making the old feature towards the new-class manifold direction or away from the new-class manifold direction. Both of them increase the variance of the old-class training feature and reduce the variance gap compared to the new classes. However, as shown in Table 6, only the variation towards new-class manifold improves the performance. Adding the new-class feature to the original feature causes the perturbed feature closer to the new-class feature manifold and produces a more informative gradient to force the original feature far away from the new-class feature. This leads to more discriminativeness between old-class features and new-class features. On the opposite, minus the new-class feature from the original feature would cause the final perturbed feature far way from

the new-class feature and causes the gradient to be less informative. This fails to take the old-class feature to be overlapped with the new-class feature. Table 6 shows that adding the new-class feature causes the better original feature to form a large angle to the new-class feature, while minus the new-class feature results in the opposite. This experiment also shows that the variation direction is important in continual learning, and we empirically verify that the perturbation direction towards the new-class feature manifold is more useful compared to the opposite direction.

### D.3 MOCA is Robust to Memory Size

There are a few memory buffer settings available in Table 2 and Table 3. To better evaluate the impact of memory size, we compare the ER baseline and our method in a wider range of memory buffer sizes. As can be seen in Table 7, both model-agnostic and model-based MOCA consistently improve the baseline under a wide range of memory buffer sizes. MOCA achieves the largest performance gain when the memory size is between 200 and 2000.

| Method | CIFAR-100 | | | |
| --- | --- | --- | --- | --- |
| | $k$=50 | $k$=200 | $k$=2000 | $k$=20000 |
| ER (Riemer et al., 2018) | 19.94 | 22.14 | 43.54 | 66.39 |
| + Gaussian | 23.56 | 27.51 | 49.61 | 67.34 |
| + WAP | **25.12** | **30.16** | **52.92** | **67.95** |

Table 7: Effect of memory buffer size to MOCA with Gaussian or WAP.

The effectiveness of MOCA will be affected when the memory buffer is extremely small or large. Small memory buffer is unable to cover representative features in the latent space, making the perturbation produced by MOCA less effective. On the other hand, the case of large memory buffer size resembles joint training, which will naturally reduce the effectiveness of MOCA. However, even in these two extreme cases, MOCA can still produce considerable performance gain.

### D.4 Convergence Stability of WAP

We discuss the convergence stability of WAP here. For all three continual learning settings in our paper, we use the same set of hyperparameters. There are two hyperparameters introduced by WAP. One is the number of updating iterations $T$ of the proxy model, and the other is the magnitude of the perturbation $\zeta$. In

| Method | $\zeta$ | | | |
| --- | --- | --- | --- | --- |
| | $\zeta$=0.1 | $\zeta$=5 | $\zeta$=10 | $\zeta$=50 |
| ER w/ WAP | 24.52 | 27.51 | **30.16** | 28.14 |

Table 8: Effect of the updating perturbation magnitude $\zeta$ for WAP.

the implementation, we fixed the updating iteration as $T = 1$ to reduce the additional training overhead, but a larger number of iterations could lead to better results. For example, if we run 2 iterations in the inner optimization, the performance of ER-WAP is 30.92%, as compared to 30.16% for the 1 inner iteration. The ablation of the hyperparameter $\zeta$ is shown in Table 8. According to the table, WAP exhibits a better performance than ER (22.14%) in a large range of hyperparameters (*e.g.*, from 0.1 to 50).

### D.5 Hyperspherical Classifier vs. Normal Classifier

**MOCA without Hyperspherical Classifier.** In this paper, we use the hyperspherical classifier for both the baseline methods and proposed methods. However, MOCA can also work without angle-based classifiers. The experimental results can be found in Table 9, which shows MOCA's effectiveness with normal classifier. However, without the normalization

| Method | CIFAR-100 | |
| --- | --- | --- |
| | $k$=200 | $k$=500 |
| ER | 22.14 | 31.02 |
| + WAP (normal classifier) | 29.33 | 39.25 |
| + WAP (hyperspherical classifier) | **30.16** | **40.24** |

Table 9: Effect of hyperspherical classifiers for WAP.

function provided by the angle-based classifiers, the feature norm would sometimes grow without control, which would occasionally cause some training instability. Moreover, the perturbation to the feature norm does not introduce useful information and only affects the learning rate, so we resort to the angle-based classifiers to eliminate the effect of feature norm perturbation.

**Effect of Hyperspherical Classifier for ER.** The choice of classifier has little influence on the performance of baseline(ER). The comparison between standard un-normalized classifiers and angle-based classifier is shown in Table 10. In LUCIR (Hou et al., 2019), the hyperspherical classifier is motivated by the following

| Method | CIFAR-10 | | CIFAR-100 | |
| --- | --- | --- | --- | --- |
| | $k$=200 | $k$=500 | $k$=200 | $k$=500 |
| ER (normal classifier) | **49.54** | **61.97** | 21.92 | 30.14 |
| ER (hyperspherical classifier) | 49.07 | 61.58 | **22.14** | **31.02** |

Table 10: Effect of hyperspherical classifier for the baseline method ER.

observation: the norm of the old classifier weight in the linear classifier is smaller than the norm of the new

classifier weight. Then LUCIR uses the hypersphere classifier to balance the classifier norm and reduce the bias in the classifier. In MOCA, the hypersphere classifiers do not have a large influence on the performance and are mostly used to stabilize the training.

### D.6 Perturbing Weight or Perturbing Feature in WAP?

WAP aims to find the closest decision boundary between the old and new classes by perturbing the weights. This serves as a good feature augmentation to prevent systematic bias towards the new class (due to extreme data imbalance). Alternatively, one may think "why not adversarially perturbing the features?". However, this does not work, because adversarial perturbation on features only considers the last-layer linear classifier and can not produce meaningful and informative augmentation for the feature encoder. In contrast, if we generate the augmentation by perturbing the weights of the feature encoder, the augmentation will take the feature manifold into consideration. To verify our intuition, we also conduct an experiment in Table 11 to demonstrate the effectiveness of adversarially perturbing the neural network weights instead of the features. To adversarially perturb the features, we use the fast gradient sign method (FGSM) (Goodfellow et al., 2014b). One can observe that adversarial perturbation on features actually hurts the performance.

| Method | CIFAR-100 | |
|---|---|---|
| | $k$=200 | $k$=500 |
| ER | 22.14 | 31.02 |
| + FGSM (Goodfellow et al., 2014b) | 21.54 | 29.87 |
| + WAP | 30.16 | 40.24 |

Table 11: The comparison of weight perturb method WAP and classical feature perturb method FGSM.

### D.7 MOCA in Different Continual Learning Settings

In the offline and online continual learning setting, the model-based MOCA variant WAP, which considers an adversarial perturbation and introduces the dependency on the model into the generation of perturbations, is empirically the best-performing method. However, in the proxy-based continual learning setting, the lack of old-class samples hinders the usage of WAP. In this case, another proposed MOCA variant VT, which considers the new-class representation structure, is the best-performing one.

### D.8 Computational Cost of Different MOCA Variants

We have recorded the running time (second) for the baseline and our two MOCA variants, Gaussian and WAP on CIFAR-100 with 200 buffer size. Experiments in Table 12 show that MOCA can improve performance by a large margin with a small training overhead. We also note that even if we use ER and train the neural network longer (with a time cost similar to ER-WAP), its performance is still around 22%. Therefore, it is generally desirable in practice that such a small training overhead is able to introduce a large gain.

| Method | Metrics | |
|---|---|---|
| | Training time (s) | Performance (%) |
| ER | **5562** | 22.14 |
| + Gaussian | 6147 | 27.51 |
| + WAP | 7109 | **30.16** |

Table 12: Training costs and final testing accuracy on CIFAR-100 for two MOCA variants (Gaussian and WAP).

### D.9 Why Does Approximating Joint Training Help?

Catastrophic forgetting is a well-known phenomenon that happens in continual learning, and in contrast, catastrophic forgetting does not exist in i.i.d. training. Such a comparison motivates us to look into what could be the gap that causes such a difference. To start with, we look into how the representation and gradient look like in continual learning (see Figure 1, Figure 2 and Figure 3), and then identify the representation and gradient collapse problem (*i.e.*, lack of variation for the memory buffer in the representation space). This is one of the most important motivations that drive us to diversify the intra-class variation to prevent the representation and gradient collapse. To well model the intra-class variation, we also draw inspiration from the joint training gradient in the design of our MOCA variants. Based on the derived dependency of the gradient, we develop both model-agnostic MOCA and model-based MOCA. Our extensive experiments on popular continual learning benchmarks verify the effectiveness of MOCA.

However, there are definitely better ways to approximate i.i.d. training and derive better continual learning methods. Our paper only demonstrates a few simple ways to approximate i.i.d. training, and we hope our method can be a good inspiration for future study.

### D.10 How to Chose the Perturbation Magnitude $\lambda$ for MOCA

The Perturbation Magnitude $\lambda$ decides the degree of intra-class representation diversification in the MOCA framework. It's influence has been shown in Figure 8. In this section, we emphasize that a appropriate $\lambda$ can not only achieve an approving performance improvements, but also can benefits the learned representations. We evaluate the learned feature representations in terms of angular fisher score. A lower angular fisher score means better discriminability of learned representations. Our

| Method | $\lambda$ | | | | |
| --- | --- | --- | --- | --- | --- |
| | $\lambda$=0 | $\lambda$=1 | $\lambda$=2 | $\lambda$=3 | $\lambda$=4 |
| ER w/ Gaussian | 1.48 | 1.29 | 1.07 | 1.08 | 1.38 |
| ER w/ WAP | 1.48 | 1.09 | 1.08 | 2.54 | Nan |

Table 13: The angular fisher score of learned feature in different perturbation magnitude $\lambda$ for two MOCA variants, Gaussian and WAP. $\lambda = 0$ means the baseline method ER.

two variants Gaussian and WAP of Moca show the best representation with $\lambda = 2$, while too small $\lambda$ would make MOCA hard to take effect, and too large $\lambda$ would also cause poor performance due to the damage to the model convergence.

### D.11 Comparison between vMF Distribution and Gaussian Distribution

According to Table 1, vMF performs worse than Gaussian in the online continual learning setting, comparable to Gaussian in the proxy-based continual learning setting, and better than Gaussian in the offline continual learning setting. In fact, vMF can directly produce perturbation on the hypersphere, which makes the magnitude of perturbation easier to control. Moreover, we have varied the hyperparameters for vMF and Gaussian in Figure 8. As one can see from Figure 8, the best performance achieved by vMF is consistently better than the best performance achieved by Gaussian. This is due to the fact that vMF distribution can easily produce effective perturbation on the hypersphere. By adjusting the concentration scale of the vMF noise, one can control the produced noise on the hypersphere. This property of vMF makes it much easier to find the best hyperparameters for MOCA.

