# OpenReview forum: "Continual Learning by Modeling Intra-Class Variation"
_TMLR — Accepted by TMLR_

### Review · Reviewer_Rind · 2022-11-04

**Summary Of Contributions:**

This paper aims to model the intra-class variation for tackling the catastrophic forgetting issue with the memory-based continual learning methods, by diversifying the representation spaces of target classes. In particular, to add variation in the representation space, the authors explore two individual directions: model-agnostic variation where the variation is made by simple parametric distributions; and model-based variation where the variation is made by characteristics of trained neural networks. The authors evaluate the proposed method, named as MOCA, on offline, online, and proxy continual learning setups with CIFAR-10, CIFAR-100, and TinyImageNet datasets, on which the proposed MOCA, which is used along with other memory-based continual learning methods as the plug-and-play component, brings performance improvements.


**Audience:**

Yes

**Broader Impact Concerns:**

While there are no explicit sections for the broader impact statement, I do not have concerns about the ethical implications of this work.

**Claims And Evidence:**

Yes

**Requested Changes:**

### Minor suggestions that would strengthen the work in my point of view
* Why i.i.d. training is necessary for continual learning, and why such the i.i.d. training scheme is achieved by adding diversity in intra-class representations are unclear, which I suggest authors to more clearly describe in the next revision.
* It would be great, if the authors experimentally justify the necessity of i.i.d. training for continual learning, and its relationship to the proposed MOCA that adds feature variations.
* I suggest authors to tone-down the claim for results in Figure 9. The authors argue the Gaussian version of the proposed MOCA is more effective in capturing the class discriminativeness, compared to the naive baseline; however, the results between them are similar.

**Strengths And Weaknesses:**

### Strengths
* The motivation of adding intra-class variation to prevent representation/gradient collapse issues is clear, which is also experimentally justified with results in Figure 1, Figure 2, and Figure 3.
* The authors extensively conduct experiments and analyses for the proposed MOCA on multiple datasets across different continual learning setups.
* The proposed MOCA, especially the weight-adversarial perturbation version, brings performance improvement, when combined with existing memory-based continual learning methods.
* This paper is very well-written, and easy-to-follow.

### Weaknesses
* The motivation on necessity of i.i.d. training for continual learning with its relation to the proposed MOCA is a bit unclear, and this point is also not experimentally justified well.

---

> ### Author Response · Authors · 2022-11-27
> **Response to Reviewer Rind**
>
> We sincerely thank the reviewer for the positive and encouraging comments. We are deeply appreciative of all the detailed suggestions as well as the recognition of our novelty. We take every raised question seriously and address them one by one. We hope that our response can clarify your concerns. If there are any other concerns, feel free to let us know and we will be more than happy to address them.
>
> ---
>
> **Q1: The motivation on necessity of i.i.d. training for continual learning with its relation to the proposed MOCA is a bit unclear, and this point is also not experimentally justified well.**
>
> A1: Great question! We derive the gradient of i.i.d. training in order to show that modeling intra-class variation in the representation space can serve as a way to approximate the joint training gradients. To well model the intra-class variation, we also draw inspiration from the joint training gradient to design our MOCA variants. For example, the dependency on the neural network weights inspires our model-based MOCA. For the experiments, we show that the model-based MOCA generally performs better than model-agnostic MOCA, verifying that approaching the joint gradients can indeed be beneficial.
>
> ---
>
> **Q2: Why i.i.d. training is necessary for continual learning, and why such the i.i.d. training scheme is achieved by adding diversity in intra-class representations are unclear, which I suggest authors to more clearly describe in the next revision.**
>
> A2: Great question! Catastrophic forgetting is a well-known phenomenon that happens in continual learning, and in contrast, catastrophic forgetting does not exist in i.i.d. training. Such a comparison motivates us to look into what could be the gap that causes such a difference. To start with, we look into how the representation and gradient look like in continual learning (see Figure 1, Figure 2 and Figure 3 in the main paper), and then identify the representation and gradient collapse problem (lack of variation for the memory buffer in representation space). This is one of the most important motivations that drive us to diversify the intra-class variation to prevent representation and gradient collapse.
>
> Thanks for the suggestion! We will improve the description and discussion for the motivation in our revision.
>
> ---
>
> **Q3: It would be great, if the authors experimentally justify the necessity of i.i.d. training for continual learning, and its relationship to the proposed MOCA that adds feature variations.**
>
> A3: Thanks for the suggestion! MOCA is motivated by bridging the gap between i.i.d. training and continual learning because i.i.d. training does not incur catastrophic forgetting. Therefore, our idea is to identify the crucial difference between i.i.d. training and continual learning, and then draw inspiration from the difference in order to improve the performance in continual learning. To this end, we conduct some experiments in Figure 1, Figure 2 and Figure 3 to demonstrate that both the representation and gradient in i.i.d. training have much larger variations than continual learning. These empirical observations serve as the initial motivation to bridge the gap between i.i.d. training and continual learning.
>
> In order to better bridge such a gap, we derive the i.i.d. training gradient and look into more specifically how to approximate i.i.d. training. Based on the derived dependency of the gradient, we develop both model-agnostic MOCA and model-based MOCA. Our extensive experiments on popular continual learning benchmarks verify the effectiveness of MOCA.
>
> However, there are definitely better ways to approximate i.i.d. training and derive better continual learning methods. Our paper only demonstrates a few simple ways to approximate i.i.d. training, and we hope our method can be a good inspiration for future study.
>
> ---
>
> **Q4: I suggest authors to tone-down the claim for results in Figure 9. The authors argue the Gaussian version of the proposed MOCA is more effective in capturing the class discriminativeness, compared to the naive baseline; however, the results between them are similar.**
>
> A4: Thanks for the suggestion! Figure 9 compares the difference among classifiers for the ER baseline, ER with Gaussian, and ER with WAP. Figure 9 elaborates that WAP can learn more discriminative classifiers than baseline and gaussian. Although our Gaussian variant may not produce significantly more discriminative classifiers, it does not indicate that the Gaussian variant does not help continual learning. The performance of continual learning is related to both the feature encoder and the classifier. The gaussian variant can still greatly increase the gradient diversity and then improve the feature encoder. Besides, the Gaussian variant still looks slightly better than the baseline for the classifier discriminativeness. To address the reviewer’s concerns, we will revise the original description in Section 5.3 by toning down the claim, and also add more discussions.

---

### Review · Reviewer_ef5C · 2022-11-11

**Summary Of Contributions:**

The authors propose methods to diversify the representations to avoid neural networks from catastrophic forgetting. On top of the thorough analyses and dissections of different variants of their method proposed, the authors provide a thorough and impressive empirical study.

**Audience:**

Yes

**Broader Impact Concerns:**

-

**Claims And Evidence:**

Yes

**Requested Changes:**

-

**Strengths And Weaknesses:**

I find the paper well written and the theoretical and empirical study very interesting and thorough. Great job! Although I can not judge the novelty of the different variants proposed, I am certain the the TMLR audience will find this paper very valuable.

Comments:
In CL, all hyperparameter will effect the models performance when trained continually (https://arxiv.org/abs/2202.00275, https://arxiv.org/pdf/2006.06958.pdf).  Can you provide one ablation study how hyperparameters (please provide them in the appendix)
effect the performance and also the representations? I think this would benefit the paper and give more intuition about how the distribution of the representations change given different hp choices.
Also for the related work section, these papers came to mind which also use replay / memory to prevent forgetting on the meta-level. https://arxiv.org/abs/1906.00695,

---

> ### Author Response · Authors · 2022-11-27
> **Response to Reviewer ef5C**
>
> We sincerely thank the reviewer for the positive and encouraging comments. We are deeply appreciative of all the detailed suggestions as well as the recognition of our novelty. We take every raised question seriously and address them one by one. We hope that our response can clarify your concerns. If there are any other concerns, feel free to let us know and we will be more than happy to address them.
>
> ---
>
> **Q1: In CL, all hyperparameters will affect the models performance when trained continually. Can you provide one ablation study how hyperparameters (please provide them in the appendix) affect the performance and also the representations?**
>
> A1: Thanks for your good suggestions! Actually, the most important hyperparameter in our framework MOCA is $\lambda$, which controls the magnitude of feature perturbations. This has been shown in Figure 8. Also, for WAP, we should select a learning rate for the proxy model. the ablation of the learning rate of the inner adversarial optimization for the proxy model is given below:
>
> |  Inner learning rate  | 0.1  | 5 | 10 | 50 |
> |  ----                 | ---- | ----  | ----  | ----  |
> | ER w/ WAP             | 24.52  | 27.51  | 30.16 | 28.14 |
>
> ---
>
> **Q2: I think this would benefit the paper and give more intuition about how the distribution of the representations change given different hp choices. Also for the related work section, these papers came to mind which also use replay / memory to prevent forgetting on the meta-level. https://arxiv.org/abs/1906.00695,**
>
> A2: Thanks for the great question! Actually, Figure 8 shows the change of modeled representation diversity according to the hyperparameter λ. It shows with a large perturbation magnitude λ, the intra-class variation of the representations goes larger, which is a verification of our motivation. Moreover, we evaluate the learned feature representations in terms of angular fisher score. A lower angular fisher score means better discriminability of learned representations. Our two variants Gaussian and WAP of Moca show the best representation with $\lambda=2$, while too small $\lambda$ would make MOCA hard to take effect, and too large $\lambda$ would also cause poor performance due to the damage to the model convergence.
>
> |  lambda  | 0 (baseline) | 1 | 2 | 3 | 4 |
> |  ----  | ---- | ----  | ----  | ----  | ----  |
> | ER w/ Gaussian| 1.48  | 1.29  | 1.07 | 1.08 | 1.38 |
> | ER w/ WAP | 1.48  | 1.09  | 1.08 | 2.54 | Nan |
>
>
> Thanks for the related papers! We have cited and discussed them in the revision.

---

### Review · Reviewer_13be · 2022-11-11

**Summary Of Contributions:**

Continual learning methods must balance between adapting to the new task while maintaining good performance on all previous tasks. This paper proposes MOCA, a classification framework that updates representations of old data to improve intra-class variation on the old classes. This uses data augmentation in representation space to create more prototype representations when training the representation model. For model-agnostic MOCA, perturbed representations are sampled according to a parametric distribution such as Gaussian or von Mises-Fisher (vMF). For model-aware MOCA, perturbations depend on the feature representation $h_\theta$ model parameters $\theta$, e.g. dropout or adversarial perturbation. Experimental results show that MOCA improves continual learning compared to several offline/online memory-based methods as well as proxy-based methods.

**Audience:**

Yes

**Broader Impact Concerns:**

No broader impact concerns.

**Claims And Evidence:**

Yes

**Requested Changes:**

### Critical
- Clarify $k$ vs $k_t$ in Section 3.1
- Explain poor performance of vMF distribution when compared to Gaussian distribution on model-agnostic MOCA
- Modify notation throughout Section 4 to clarify where mean representations would not work
- Summarize empirical findings by outlining which MOCA variant is most appropriate in which learning setting

### Non-critical
- Compare/discuss computational cost of the different MOCA variants (e.g. running time, memory)
- Comment on Figure 1: for model-agnostic MOCA intra class variation is actually larger for old classes than for new classes. Does this affect performance?
- In addition to Section 5.5, mention manifold mixup earlier in the introduction when discussing perturbation in the representation space
- Justify the assumption for MOCA VT that "intra-class representation variations for different classes are similar"

### Typos
- "pastly learned parameters" should be "previously learned parameters"
- "More additional experiments" should be "Additional experiments"


**Strengths And Weaknesses:**

### Strengths
- Interesting and novel approach to an important problem
- Orthogonal/compatible with other methods for continual learning
- Impressive performance improvements when combined with SoTA baselines

### Weaknesses
- Clarity
	- Notation in Section 3.1 alternates between total number of classes $k$ and classes for the $t$-th task $k_t$. Must the total number of classes be known in advance?
	- In Section 4.3, model-based MOCA is separated into variants that require prototypes (DOA-old and WAP) and variants that can work with mean representations (DOA-new and VT). However, $h_\theta(x^{old})$ notation is used throughout, which suggests that all models use $x^{old}$. This notation can be made clearer by emphasizing where $h_\theta(x^{old})$ is necessary and mean representations $\bar{f}$ would not work.
	- In Section 5.3, the angle between classifiers is used without definition for evaluating classifiers
- vMF distribution is well-motivated by previous works on modeling intra-class variation on a hypersphere, which the authors cite. However, the performance of model-agnostic MOCA with vMF sampling is worse than with Gaussian sampling. The authors should provide an explanation or support this claim with additional experiments beyond Table 1.

---

> ### Author Response · Authors · 2022-11-27
> **Response to Reviewer 13be (Part 1)**
>
> We sincerely thank the reviewer for the positive and constructive comments. We are deeply appreciative of all the detailed suggestions as well as the recognition of our novelty. We take every raised question seriously and address them one by one. We hope that our response can clarify your concerns. If there are any other concerns, feel free to let us know and we will be more than happy to address them.
>
> ---
>
> **Q1: Notation in Section 3.1 alternates between total number of classes $k$ and classes for the k-th task $k_t$ . Must the total number of classes be known in advance?**
>
> A1: Thanks for your suggestion! We have added a description of $k$ in our revised paper. MOCA does not need to know the total number of classes. It can work with any number of classes and does not require this information in advance.
>
> ---
>
> **Q2: In Section 4.3, model-based MOCA is separated into variants that require prototypes (DOA-old and WAP) and variants that can work with mean representations (DOA-new and VT). However,  notation is used throughout, which suggests that all models use. This notation can be made clearer by emphasizing where is necessary and mean representations would not work.**
>
> A2: Thanks for the suggestion! The motivation of MOCA is to diversify the old-class representations in the memory buffer. Hence, in our paper, we use the $x^{old}$ throughout the paper. Since we aim to diversify the old class intra-class representations, all the MOCA variants will typically use the original $x^{old}$ to produce the feature $h_{\theta}(x^{old})$ in memory-based continual learning settings (offline continual learning & online continual learning). In the proxy-based continual learning, we don’t have access to $x^{old}$, but the old class-mean representations $\bar{f}$ are stored and re-usable (according to the setting). Therefore, DOA-old and WAP will not work in this setting. However, DOA and VT are not affected as they are only dependent on new class samples to produce perturbations and do not need $x^{old}$. We are sorry if our paper is not clear enough and causes some confusion. We have revised the description of proxy-based continual learning in our paper, where class-mean representations are necessary to represent old classes.
>
> ---
>
> **Q3: In Section 5.3, the angle between classifiers is used without definition for evaluating classifiers**
>
> A3: Thanks for pointing it out! In section 5.3, the angle between classes is defined as the average angle between each learned classifier. Because the learned classifier vector can be approximately viewed as the class center of each class, visualizing the angle between classifiers can be a good measurement to evaluate the learned representations. We will include all the necessary definitions in the revision.
>
> ---
>
> **Q4: vMF distribution is well-motivated by previous works on modeling intra-class variation on a hypersphere, which the authors cite. However, the performance of model-agnostic MOCA with vMF sampling is worse than with Gaussian sampling. The authors should provide an explanation or support this claim with additional experiments beyond Table 1.**
>
> A4: Thanks for the suggestion! We believe that both MOCA variants (vMF and Gaussian) actually have similar performance. According to Table 1, vMF performs worse than Gaussian in the online continual learning setting, comparable to Gaussian in the proxy-based continual learning setting, and better than Gaussian in the offline continual learning setting. In fact, vMF can directly produce perturbation on the hypersphere, which makes the magnitude of perturbation easier to control. Moreover, we have varied the hyperparameters for vMF and Gaussian in Figure 8. As one can see from Figure 8, the best performance achieved by vMF is consistently better than the best performance achieved by Gaussian. We will also include more discussions to compare vMF and Gaussian.
>
> ---
>
> **Q5: Clarify $k$ vs $k_t$ in Section 3.1**
>
> A5: Thanks for the suggestion! We are sorry that $k$ and $k_t$ may not be defined in a clear way. $k_t$ denotes the $k_t$-th class, which is just the class index. For the classes between $k_{t-1}+1$ to $k_t$, they belong to the $t$-th continual task. $k$ is the total number of classes. We have improved the descriptions in the revised paper.
>
> ---
>
> **Q6: Explain poor performance of vMF distribution when compared to Gaussian distribution on model-agnostic MOCA**
>
> A6: Thanks for the suggestion! Please refer to our response in A4.
>
> ---
>
> **Q7: Modify notation throughout Section 4 to clarify where mean representations would not work**
>
> A7: Thanks for the suggestion! Please refer to our response in A3.

---

> ### Author Response · Authors · 2022-11-27
> **Response to Reviewer 13be (Part 2)**
>
> **Q8: Summarize empirical findings by outlining which MOCA variant is most appropriate in which learning setting**
>
> A8: Good question! In both the offline continual learning setting and online continual learning setting, the model-based MOCA variant WAP, which considers an adversarial perturbation and introduces the dependency on the model into the generation of perturbations, is empirically the best-performing method. However, in the proxy-based continual learning setting, the lack of old-class samples hinders the usage of WAP. In this case, another proposed MOCA variant VT, which considers the new-class representation structure, is the best-performing one. We will add a summarization to the paper.
>
> ---
>
> **Q9: Compare/discuss computational cost of the different MOCA variants (e.g. running time, memory)**
>
> A9: Good question! We have recorded the running time (/s) for the baseline and our two MOCA variants, Gaussian and WAP on CIFAR-100. Experiments show that MOCA can improve performance by a large margin with a small training overhead. We also note that even if we use ER and train the neural network longer (with a time cost similar to ER-WAP), its performance is still around 22%. Therefore, it is generally beneficial that such a small training overhead is able to introduce a large performance gain.
>
> |                       | Training Time (s)  | Performance (%)|
> |  ----                 | ---- | ---- |
> | ER                    | 5562  | 22.14 |
> | ER w/ Gaussian        | 6147  | 27,51 |
> | ER w/ WAP             | 7109  | 30.16 |
>
> ---
>
> **Q10: Comment on Figure 1: for model-agnostic MOCA intra class variation is actually larger for old classes than for new classes. Does this affect performance?**
>
> A10: Good question! Figure 8 actually implies the answer to the question. A larger variation would decrease the performance gain, which indicates that it is not always better to have a larger variation (hence larger intra-class variance) and there exists a suitable variation range. In Figure 8, we show that an extremely large perturbation magnitude $\lambda$ would lead to a poor performance.
>
> ---
>
> **Q11: In addition to Section 5.5, mention manifold mixup earlier in the introduction when discussing perturbation in the representation space**
>
> A11: Good suggestion! We will revise our paper accordingly.
>
> ---
>
> **Q12: Justify the assumption for MOCA VT that "intra-class representation variations for different classes are similar"**
>
> A12: Great question! The assumption that intra-class representations are similar comes from the visualization of deeply learned features (as shown in the anonymous figure below). While it is hard to justify this assumption from a theoretical perspective, this assumption is empirically valid from the visualization. We can understand the correctness of this idea from the figure below. This figure shows 10 classes trained with CE loss in the joint training. According to this figure, we can draw this conclusion easily: The intra-class representation variations for different classes are similar, although the class mean of different classes is distributed in different locations.
>
> Despite the empirical nature of this assumption, we find that VT works reasonably well in practice, which partially justifies its usefulness as an assumption.
>
> https://anonymous.4open.science/r/rebuttal-7D75/fig2_00.png
>
> ---
>
> **Q13: Typos**
>
> A13: Thanks for pointing them out! We have fixed them in the revision.

---

### Review · Reviewer_ft7X · 2022-11-15

**Summary Of Contributions:**

The authors investigate the evolution of representation in continual and offline learning models and find a major difference in the way features organize. On this basis, they propose a set of methods aimed at ensuring that representations for past classes do not collapse by taking intra-class variation as a key objective in learning. The authors show that their proposed approach is rather effective when combined on top of SOTA approaches on a variety of benchmarks and highlight interesting properties of their proposal by means of additional ablation studies.

**Audience:**

Yes

**Claims And Evidence:**

Yes

**Requested Changes:**

I suggest the following changes:

+ introducing a discussion on the memory impact of training with the different MOCA versions;
+ detail whether WAP is particularly hard to train or not;
+ motivate whether the choice of adopting an angle-based classifier has an impact on accuracy of the evaluated models and how this choice is related to LUCIR and/or other CL literature.

**Strengths And Weaknesses:**

I found the paper to be overall interesting, well written and easy to follow, for which I congratulate the authors.

Strengths:

+ This paper is well motivated and the proposal of MOCA starts from a clearly-exposed problem affecting the dynamics of learning. For this reason, the exposition of the model is easy to follow and understand.
+ I find the proposed approach to be rather unique in the landscape of current CL methods, in that it acts as an open-ended regularizer in function space that not only pushes representations to be coherent with the past (a la LwF/iCaRL), but also endows them with desirable a-priori properties by comparing the representation-space behavior of different models.
+ The presented ablation studies are quite interesting and successfully highlight the effectiveness of the proposed MOCA

Weaknesses:

+ The authors are not clear on the impact on memory of the proposed methods. What is the memory impact of training with MOCA compared with the baseline methods?
+ The authors provide a clear indication as to what version of MOCA performs best, i.e. WAP. However, given its adversarial nature, I would like there to be more clarity on details of its training procedure and whether its training is stable and converges easily.
+ Some of the presented information might are not clearly motivated and might need further details, some examples:
    + on page 9 (5 - experimental setting) the authors write that they provide angle-based classifiers for all evaluated method. Does this choice have any consequence on competitors that are not originally designed to work under this assumption?
    + on page 9 (top) wap is said to perturb the network weight adversarially, making it good at preventing catastrophic forgetting. This is not really an obvious consequence of adversarial training. How are these two facts related?
+ The experiments on the online setting show several competitor methods (SS-IL, iCaRL) outperforming some WAP-enhanced baselines. Could the authors try and combine MOCA with SS-IL or iCaRL? Is MOCA effective when applied on top of these methods?
+ The choice of the hypersphere classifier might be somewhat related with the deign choice applied in LUCIR and other methods: a reader could find it beneficial to discuss the similarities and differences with this approach.
+ I believe that the use of the term *prototypes* to indicate examples in the replay buffer is not correct. Prototype generally indicates points in representation space that are a model for other latent point of the same class.

---

> ### Author Response · Authors · 2022-11-27
> **Response to Reviewer ft7X (Part 1)**
>
> We sincerely thank the reviewer for the positive and constructive comments. We are deeply appreciative of all the detailed suggestions as well as the recognition of our novelty. We take every raised question seriously and address them one by one. We hope that our response can clarify your concerns. If there are any other concerns, feel free to let us know and we will be more than happy to address them.
>
> ---
>
> **Q1: The authors are not clear on the impact on memory of the proposed methods. What is the memory impact of training with MOCA compared with the baseline methods?**
>
> A1: Good question! There are a few memory buffer settings available in Table 2&3. To better evaluate the impact of memory, we compare the ER baseline and our method in a wider range of memory buffer sizes. As can be seen in the table below, both model-agnostic and model-based MOCA consistently improve the baseline under a wide range of memory buffer sizes. MOCA achieves the largest performance gain when the memory size is between 200 to 2000. The effectiveness of MOCA will be affected when the memory buffer is extremely small or large. Small memory buffer is unable to cover representative features in the latent space, making the perturbation produced by MOCA less effective. On the other hand, the case of large memory buffer size resembles joint training, which will naturally reduce the effectiveness of MOCA. However, even in these two extreme cases, MOCA can still produce considerable performance gain.
>
> |  Memory size      | 50  | 200 | 2000 | 20000 |
> |  ----             | ---- | ----  | ----  | ----  |
> | ER                | 19.94  | 22.14  | 43.54 | 66.39 |
> | ER w/ Gaussian    | 23.56  | 27.51  | 49.61 | 67.34 |
> | ER w/ WAP         | 25.12  | 30.16  | 52.92 | 67.95 |
>
> ---
>
> **Q2: The authors provide a clear indication as to what version of MOCA performs best, i.e. WAP. However, given its adversarial nature, I would like there to be more clarity on details of its training procedure and whether its training is stable and converges easily.**
>
> A2: Thanks for the suggestion! We will put the pseudo-code as well as the training details for WAP in the appendix, and hopefully it can improve the clarity of WAP. Moreover, our source code for reproducing the results will be made available.
>
> In each iteration, WAP first initializes a proxy model $\theta_{adv}$ which has the same parameters as the current learned model ${\theta}$. Then, old-class data from $\mathcal{M}$ would be fed to the proxy model $\theta_{adv}$. To diversify the intra-class representation of old classes, the proxy model would be expected to produce representations that differ from the original representations. In WAP, the proxy model is adversarially updated such that it will tend to classify the old-class sample to the new class. By doing this, we get the adversarially updated proxy model $\theta_{adv}=\theta+\Delta\theta$. Then, the produced feature by the updated proxy model would be regarded as the effective feature perturbations. In each training iteration, after getting these perturbations, WAP adds the perturbation to the original feature produced by the original model $\theta$ for old-class samples, achieving the goal of intra-class representation diversification. This diversification process would eventually affect the training of the original model $\theta$.
> The convergence of WAP is quite stable. For all the 3 continual learning settings in our paper, we use the same set of hyperparameters. There are two hyperparameters introduced by WAP. One is the number of updating iterations of the proxy model, and the other is the magnitude of the perturbation $\zeta$. In the implementation, we fixed the updating iteration as 1 for a low training cost, but a larger number of iterations could lead to better results. For example, if we run 2 iterations in the inner optimization, the performance of ER-WAP is 30.92%, as compared to 30.16% for the 1 inner iteration.
>
> The ablation of the hyperparameter $\zeta$ is shown below. According to the table, WAP can show a better performance than ER(22.14) in a large range of hyperparameters (0.1-50)
>
> |  zeta  	| 0.1  | 5 | 10 | 50 |
> |  ----               | ---- | ----  | ----  | ----  |
> | ER w/ WAP | 24.52  | 27.51  | 30.16 | 28.14 |

---

> ### Author Response · Authors · 2022-11-27
> **Response to Reviewer ft7X (Part 2)**
>
> **Q3: Some of the presented information might are not clearly motivated and might need further details, some examples:
> Q3.1: on page 9 (5 - experimental setting) the authors write that they provide angle-based classifiers for all evaluated method. Does this choice have any consequence on competitors that are not originally designed to work under this assumption?**
>
> Thanks for the suggestion. We will improve the clarity of our paper following the suggestion. To address the reviewer’s concerns, we provide additional information for both Q3.1 and Q3.2.
>
> A3.1: First of all, our method can also work well without angle-based classifiers. The experimental results can be found below. However, we do find that, without the normalization function provided by the angle-based classifiers, the feature norm would sometimes grow without control, which would occasionally cause some training instability. Moreover, the perturbation to the feature norm does not introduce useful information and only affects the learning rate, so we resort to the angle-based classifiers to eliminate the effect of feature norm perturbation.
>
> |                       | cifar-100 (200) | cifar-100 (500) |
> |  ----                 | ---- | ----  |
> | ER                    | 22.14 | 31.02  |
> | ER w/ WAP (normal classifier)                 | 29.33  | 39.25  |
> | ER w/ WAP (hypersphere classifier)            | 30.16  | 40.24  |
>
> Second, this choice of classifiers also has little influence on the performance of baseline (ER). The comparison between standard classifiers and angle-based classifiers is shown below:
>
> | |  cifar-10 (200) | cifar-100 (200) | cifar-10 (500) | cifar-100 (500)|
> |  ----                 | ---- | ----  | ----  | ----  |
> | ER (standard classifier)            | 49.54  | 21.92  | 61.97 | 30.14 |
> | ER (angle-based classifier)       | 49.07  | 22.14  | 61.58 | 31.02 |
>
>
> **Q3.2: on page 9 (top) wap is said to perturb the network weight adversarially, making it good at preventing catastrophic forgetting. This is not really an obvious consequence of adversarial training. How are these two facts related?**
>
> A3.2: Good question! We should have been more clear about this. In continual learning, catastrophic forgetting usually happens when the  new-class data comes in, and in this scenario, the weights of the neural network will be updated such that it tends to predict all the samples as the new class. To prevent such a weight update that leads to catastrophic forgetting, how should we design the feature perturbation? WAP is motivated in such a scenario. Adversarially perturbing network weights towards the new-class label aims to simulate the weight update in the catastrophic forgetting scenario. Producing the feature augmentation under the adversarially perturbed weights can prevent the neural network from such a weight update because the cross-entropy loss will push the prediction of the augmented feature to approach the old class (which is different from the adversarial direction).
>
> From a different perspective, WAP aims to find the closest decision boundary between the old and new classes by perturbing the weights. This serves as a good feature augmentation to prevent systematic bias towards the new class (due to extreme data imbalance). Alternatively, one may think “why not adversarially perturbing the features?”. However, this does not work, because adversarial perturbation on features only considers the last-layer linear classifier and may not produce meaningful and informative augmentation for the feature encoder. In contrast, if we generate the augmentation by perturbing the weights of the feature encoder, the augmentation will take the feature manifold into consideration. To verify our intuition, we also conduct an experiment to demonstrate the effectiveness of adversarially perturbing the neural network weights instead of the features. To adversarially perturb the features, we use fast gradient sign method (FGSM). One can observe that adversarial perturbation on features actually hurt the performance quite a bit.
>
> |                       | cifar-100 (200) | cifar-100 (500) |
> |  ----                 | ---- | ----  |
> | ER                    | 22.14 | 31.02  |
> | ER w/ FGSM               | 21.54  | 29.87  |
> | ER w/ WAP           | 30.16  | 40.24  |

---

> ### Author Response · Authors · 2022-11-27
> **Response to Reviewer ft7X (Part 3)**
>
> **Q4: The experiments in the online setting show several competitor methods (SS-IL, iCaRL) outperforming some WAP-enhanced baselines. Could the authors try and combine MOCA with SS-IL or iCaRL? Is MOCA effective when applied on top of these methods?**
>
> A4: Great suggestion! MOCA can be easily used in SS-IL or iCaRL. Here, we take SS-IL as an example. We find that MOCA can also improve the performance of SS-IL in the online continual learning setting. Experimental results are given below. The number in the bracket denotes the memory buffer size
>
> | |  cifar-10 (20) | cifar-10 (100) | cifar-100 (20) | cifar-100 (100)|
> |  ----                 | ---- | ----  | ----  | ----  |
> | SS-IL              | 35.54  | 42.78  | 16.20 | 26.24 |
> | SS-IL w/ WAP       | 37.21  | 43.95  | 17.54 | 27.12 |
>
> ---
>
> **Q5: The choice of the hypersphere classifier might be somewhat related with the design choice applied in LUCIR and other methods: a reader could find it beneficial to discuss the similarities and differences with this approach.**
>
> A5: Thanks for the suggestion! We agree with the reviewer that the discussion on the hypersphere classifiers could be of broad interest, and the connection/difference to LUCIR is worth discussing. In LUCIR, the hypersphere classifier is motivated by the following observation: the norm of the old classifier weight in the linear classifier is smaller than the norm of the new classifier weight. Then LUCIR uses the hypersphere classifier to balance the classifier norm and reduce the bias in the classifier. In MOCA, the hypersphere classifiers do not have a large influence on the performance and are mostly to stabilize the training.
>
> ---
>
> **Q6: I believe that the use of the term prototypes to indicate examples in the replay buffer is not correct. Prototype generally indicates points in representation space that are a model for other latent point of the same class.**
>
> A6: Thanks for your suggestion! We agree with the reviewer that prototypes are usually used to indicate points in the representation space. However, since MOCA aims at modeling the feature space, we utilize the term prototype to represent the data point itself in the memory buffer (which can be in either the input space or feature space depending on the context). We will add a footnote to make this clear to avoid any potential confusion.

---

### Author Response · Authors · 2022-11-27
**Response Summary**

Dear Reviewers and AC,

We sincerely thank all the reviewers and AC for spending time on our submission. We are deeply appreciative of all the constructive commeents and have addressed all the raised concerns. We are excited that all the reviewers find our paper novel and our contribution significant. We will substantially improve our paper following the reviewers' suggestions.

**If there are any other concerns, please feel free to let us know and we will be more than happy to address them.**

PS: ~~Our revised paper will be uploaded in 1-2 days. Sorry for the inconvenience.~~ Our revised paper has been uploaded!

Best,

Authors

---

### Author Response · Authors · 2023-01-30
**Summary of Camera Ready Version**

We sincerely thank the action editor and all the reviewers for spending the efforts and time to improve our paper. In our camera ready revision, we make the following changes:

- We included missing training details in Appendix A (Implementation Details).
- We gave extensive discussion and comparison for the angle-based classifier in Appendix D (Additional Experimental Results and Discussions).
- We added the convergence experiments and the discussions regarding WAP in Appendix D4.
- Finally, we open-sourced our PyTorch implementation for full reproducibility. The GitHub link is provided in the paper.

Again, we are deeply thankful for all the great suggestions from the action editor and reviewers. We are also happy to see our response and updated paper clarify all the potential concerns.

---

### Decision · Action_Editors · 2022-12-29

**Recommendation:** Accept with minor revision

**Comment:**

This paper models the intra-class variation for tackling the catastrophic forgetting issue with the memory-based continual learning methods, by diversifying the representation spaces of target classes. Given the original submission, the reviewers posted quite some comments, but they are well addressed by the author response and the revised version of their paper. One reviewer suggested that the authors should make one more effort to clarify the missing details on training, such as the convergence and the angle-based classifiers. We believe by making such minor revisions, the paper quality will be further improved.

**Audience:**

Yes. Continual learning is a recent trending topic, and the TMLR audience will be interested in this paper.

**Claims And Evidence:**

The paper is in a good shape, providing supports on most of the claims.